

# Dust semi-direct effects: Dust-induced longwave radiation influences low-level cloud response to free-tropospheric dust over the North Atlantic Ocean.

Satyendra K. Pandey and Adeyemi A. Adebiyi

*Department of Life and Environmental Sciences, University of California - Merced, Merced, CA, USA*

*Correspondence to*: Satyendra K. Pandey ([spandey9@ucmerced.edu](mailto:spandey9@ucmerced.edu))

## Abstract

Aerosol semi-direct effect is the adjustment of the radiative budget due to the cloud response to radiation absorption. Although dust accounts for about a third of aerosols' shortwave absorption, our understanding of its semi-direct effect often relies on traditional shortwave-focused mechanisms previously established for biomass-burning aerosols, and implications of dust longwave absorption on clouds have yet to be explored. Here, we assess the low-level cloud cover (LLCC) response to changes in properties and

characteristics of the free-tropospheric dust layer over the North Atlantic Ocean (May-August, 2007-2017). We find that, consistent with previous studies, LLCC typically responds positively (increases in clouds) to an overlying dust layer. However, this response weakens with increasing dust optical depth (DOD), geometric thickness (GT), and dust-layer base (DB). Specifically, we find that the LLCC response weakens by 4.3±1.04% and 1.6±0.65%, respectively, for a one-standard-deviation increase in DOD and

GT, and a smaller response to DB (0.19±0.45%). We also find that the weakened LLCC response is primarily due to enhanced dust-induced longwave-dominated cloud-top warming, which counteracts the mean cloud-top cooling by as much as 19% (mean of 9%). Sensitivity analysis further indicates that the variability in dust properties, influenced by dust size distribution and refractive index, dominates the changes in dust-induced cloud-top warming, rather than variabilities in cloud properties or

thermodynamic profiles. Our result adds to the traditional understanding of LLCC enhancement through shortwave-driven atmospheric stability, often associated with aerosol semi-direct effects, and highlights the role of dust-induced cloud-top longwave warming in dust semi-direct effects.

**Keywords:** Dust; Longwave semi-direct effect; Cloud-top radiative cooling; Low-level clouds

## 1 Introduction

Atmospheric aerosols remain a complex but important component of the Earth's climate system, representing a major source of uncertainties in our understanding of the current climate and its future projections (Bellouin, 2015; Bellouin et al., 2020; Bender, 2020; Forster et al., 2023). This is because





aerosols can interact with the climate system through several complex pathways, influencing the overall radiative budget at the surface and top of the atmosphere (Bellouin et al., 2020; Li et al., 2022; Satheesh and Krishnamoorthy, 2005). For example, aerosols can directly impact the climate by scattering and absorbing solar and terrestrial radiation (Atwater, 1970; Chlek and Coakley, 1974). It can also impact the climate indirectly by serving as cloud condensation nuclei or ice nuclei, influencing cloud formation, reflectivity, and lifetime (Albrecht, 1989; Twomey, 1974). Another way the aerosols influence climate without contributing to cloud formation is by modulating the thermodynamic profiles of the atmosphere where the clouds exist (Allen et al., 2019; Amiri-Farahani et al., 2017; Baró Pérez et al., 2021; Hansen et al., 1997; Herbert et al., 2020; Johnson et al., 2004; Koren et al., 2004; Lu et al., 2021). This latter pathway of aerosol-climate interactions called aerosol semi-direct effect (SDE), is the adjustment of the radiative budget at the top of the atmosphere due to the cloud response to aerosol radiation absorption (Hansen et al., 1997; Johnson et al., 2004; Koren et al., 2004). Previous studies have suggested that aerosol semi-direct effects could warm or cool the climate system, depending on the relative vertical position of the aerosol and cloud layers (Koch and Del Genio, 2010). However, our understanding of the exact processes that establish this aerosol semi-direct effect remains very limited, contributing a substantial fraction to the uncertainties associated with aerosol-radiation interactions and, consequently, the overall radiative budget at the top of the atmosphere (Adebiyi et al., 2023; Bellouin et al., 2020; Forster et al., 2023; Koch and Del Genio, 2010). Further, it remains unclear whether these processes related to aerosol semi-direct effects are consistent across the different absorbing aerosols in the atmosphere, including biomass-burning and mineral dust aerosols, and to what degree they may differ.

Most previous studies estimating aerosol semi-direct effects have largely focused on biomass-burning smoke aerosols, including black carbon, that dominate absorbing aerosols in the atmosphere, and these studies have shaped our general understanding of processes associated with SDE (Bond et al., 2013; Jacobson, 2014; Koch and Del Genio, 2010; Mallet et al., 2020). Specifically, these studies have indicated that the semi-direct effects of biomass-burning aerosols and the corresponding cloud response depend on cloud types, aerosol properties, and the relative position of the aerosol and cloud layers (Haywood et al., 2020; Herbert and Stier, 2023; Wilcox, 2010). For example, when biomass-burning aerosols are within a cloud layer, the warming due to shortwave absorption can '*burn off*' the cloud, decreasing the cloud cover, which produces a positive semi-direct effect (Johnson et al., 2004; Koren et al., 2004). In addition, when the biomass-burning aerosols are distinctly above the cloud layer, the shortwave warming can increase the lower tropospheric stability, increasing the moisture build-up in the boundary layer and resulting in increased cloud cover and a negative semi-direct effect (Wilcox, 2012). Although semi-direct effects from biomass-burning aerosols occur for all cloud types, their influences on marine low-level clouds are far more substantial (Koch and Del Genio, 2010), dominating the aerosol-cloud processes and, therefore, the overall aerosol radiative effects over the Southeast Atlantic Ocean (Zuidema et al., 2016). Globally, low-level clouds are widespread and constitute about 40 % of all overall cloud occurrences (Schneider et al.,




2013; Stubenrauch et al., 2013), with single-layer low-level clouds, such as stratocumulus clouds, covering about one-fifth of the global ocean (Hahn and Warren, 2007). Given the widespread presence of marine low-level clouds co-existing with other absorbing aerosols, our understanding of the aerosol semi-direct effect that relies on processes associated with biomass-burning smoke aerosols alone is, therefore, incomplete.

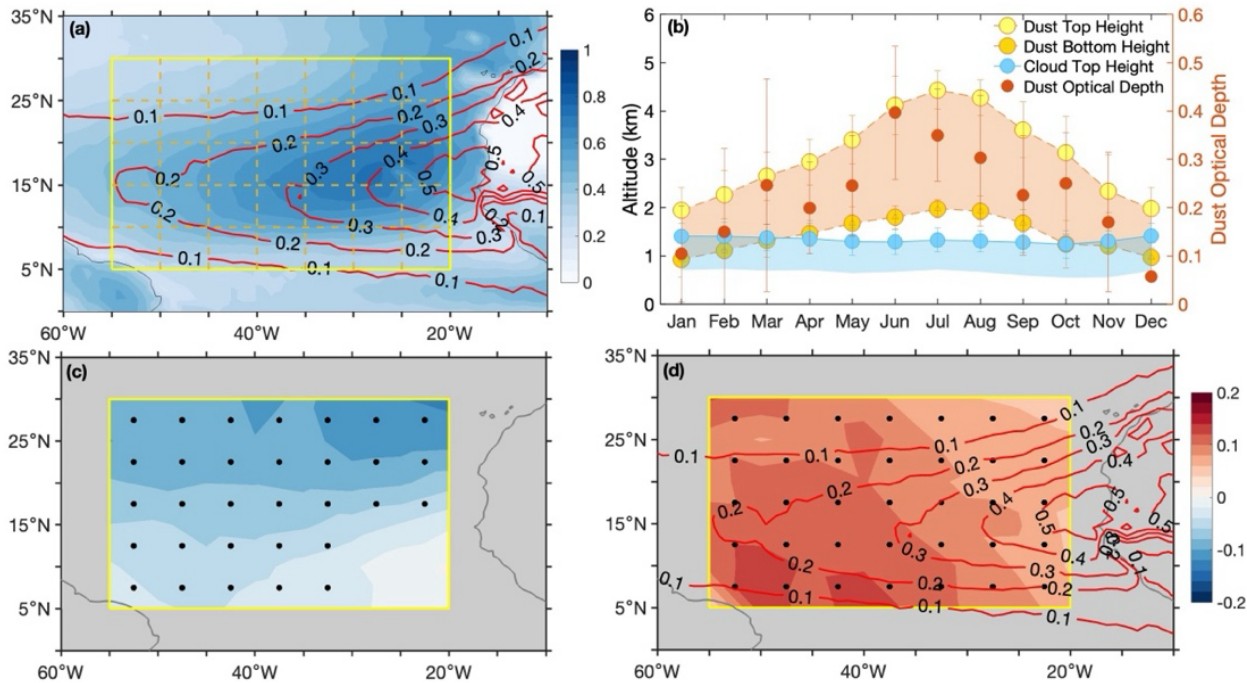

**Figure 1: (a) The spatial distribution of low-level cloud fraction (shaded contour) and column dust optical depth (red line contours). The yellow box represents the analysis domain, which consists of 5°×5° grid boxes used for categorization in our analysis (see Method). (b) The annual cycle of dust and cloud layer top and base height averaged over the analysis domain (yellow box in (a)). (c) & (d) Changes in low-level cloud fraction relative to a long-time mean over the region when (c) all aerosols and (d) only dust are present above the cloud layer. Black dots show locations that are significant at the 95 % confidence level with a two-tailed Student t-test. The cloud fraction was taken from MODIS Aqua and dust optical depth from CALIPSO Level 2 datasets between May and August 2007 and 2017.**

Although most studies shaping our understanding of aerosol semi-direct effects focus on biomass-burning smoke aerosols, few studies have examined semi-direct effects for mineral dust aerosols – another major absorbing aerosol that accounts for about 30% of the shortwave absorption in the atmosphere (Sand et al., 2021). Moreover, mineral dust also accounts for about two-thirds of all aerosol mass in the atmosphere, with more than 50 % emitted from the North African Deserts and transported westward over the North




Atlantic Ocean in what is called the Sahara Air Layer (SAL) (Adebiyi et al., 2025; Kok et al., 2021a;
Prospero et al., 2021). Over the ocean, these North African dust aerosols interact with one of the major
low-level cloud regions (Fig. 1a), influencing the radiative budget at the top of the atmosphere (Wood,
2012). During the spring and summer, the SAL has an average separation from the underlying low-level
clouds of about 0.12 km that maximizes in July (Fig. 1b). In contrast, during the winter, the dust layer is
generally transported at a lower altitude, allowing the dust particles to mix with the low-level clouds.
Previous studies have suggested that dust SDE and the associated low-level cloud responses are similar
to those described by biomass-burning aerosols (Amiri-Farahani et al., 2017; DeFlorio et al., 2014;
Doherty and Evan, 2014; Huang et al., 2006). That is, when dust aerosols are within the low-level cloud
layer, dust SDE is positive with a decrease in low-level cloud cover, while when dust is above the cloud
layer, dust SDE is negative with an increase in cloud cover. Over the North Atlantic Ocean, the
enhancement in low-level cloud cover is generally positive during the summer when the dust is above the
cloud (about +8% increase for the domain shown in Fig. 1c) compared to when other aerosols are above
the cloud (-6% decrease for the same domain region shown in Fig. 1d). In addition, like biomass-burning
aerosols, these previous studies have linked the enhancement of low-level cloudiness and the associated
negative dust SDE to the above-cloud dust layer to stronger lower-tropospheric inversion caused by dust-
induced shortwave heating near the cloud top (DeFlorio et al., 2014; Doherty and Evan, 2014).

However, unlike biomass-burning aerosols, atmospheric dust contains a significant fraction of coarser
particles (diameter ≥ 2.5 µm) that can interact with longwave radiation and, therefore, impact dust semi-
direct effect differently from other absorbing aerosols (Adebiyi et al., 2023; Adebiyi and Kok, 2020; Kok
et al., 2017, 2021b). Specifically, previous studies leveraging in-situ dust measurements have shown that
dust particles larger than 5 µm are about four times more abundant in the atmosphere than previously
estimated by climate models (Adebiyi and Kok, 2020). Furthermore, observations also showed that these
coarse dust aerosols can have sizes up to about 60 µm in the atmosphere, making them an important
component that cannot be ignored in estimating dust radiative impacts (Adebiyi et al., 2023; Ryder et al.,
2013, 2018, 2019; Weinzierl et al., 2017). Because coarse-mode dust absorbs and scatters more longwave
radiation than fine-mode dust, it can significantly cool the dust layer in the longwave, counteracting the
warming effect of shortwave radiation and, therefore, may influence the overall dust semi-direct effect
and associated low-level cloud response (Kok et al., 2023). In addition, dust-induced longwave radiation
can be emitted towards the low-level clouds, potentially impacting the cloud-top processes that deviate
from the traditional process of SDE by biomass-burning aerosols.

Furthermore, this potential dust-induced longwave influence on dust's semi-direct effect and its low-level
cloud response could depend on the properties and characteristics of the dust layer. One parameter that
can influence the dust-induced longwave absorption within the dust layer and the associated downwelling
radiation is the dust extinction or optical depth, which, in turn, depends on the dust size distribution and



dust complex refractive index (Gkikas et al., 2022; Zheng et al., 2023). For example, the vertical
distribution of coarse dust within the dust layer can affect the vertical distribution of dust absorption, the
dust-layer longwave cooling, and the magnitude of downwelling radiation, thereby influencing potential
interactions with underlying low-level clouds. In addition, for the same optical properties, the altitude and
geometric thickness of the dust layer can also influence the amount of radiation reaching the cloud top.
These inferences were recently made for biomass-burning smoke aerosols by Herbert et al. (2020), who
used large eddy simulation to show the sensitivity of the cloud properties and the resulting semi-direct
effect to the variability in the altitude and layer characteristics of the smoke overlying the low-level cloud
over the Southeast Atlantic Ocean. While this assessment was conducted using model simulation and only
for biomass-burning aerosols, it remains unclear, however, exactly how observed variabilities in optical
properties and dust-layer characteristics, including the dust layer altitude, its geometric thickness, and
separating distance, will affect underlying low-level clouds, its response to radiative warming or cooling,
and ultimately dust semi-direct effect.

To fill this gap, this study seeks to understand how the above-cloud Sahara air layer influences the dust
semi-direct effect over the North Atlantic Ocean by examining the impacts of dust optical depth and dust-
layer characteristics on underlying low-level cloud response using satellite observations. Specifically, we
leveraged aerosols and cloud information from CALIOP (Cloud-Aerosol Lidar with Orthogonal
Polarization) onboard CALIPSO (Cloud-Aerosol Lidar and Infrared Pathfinder Satellite Observation)
satellite and cloud profiling radar (CPR) aboard CloudSat. We hypothesize that the influence of above-
cloud dust on low-level cloudiness varies depending on the characteristics of the dust layer, specifically
the dust-base altitude and geometric thickness. Whereas the dust-base altitude (DB) characterizes the
proximity of the dust and cloud layers, the geometrical thickness (GT) represents the vertical extent of
the dust layer. Our results show a reduction in mean cloud cover with an increase in dust geometrical
thickness, regardless of dust-base heights, whereas cloud response to dust-base height depends on the
thickness of the layer. Additionally, with constant base altitude and geometrical thickness, our results also
show a negative cloud cover sensitivity with increasing dust optical depth that depends on the dust size
distribution and the complex refractive index of the dust layer. We find that these responses in cloud cover
to changes in dust-layer properties and characteristics primarily occur through dust modulation of the
longwave fluxes reaching the cloud top, which reduces the mean cloud-top cooling.





## 2 Data and Methods

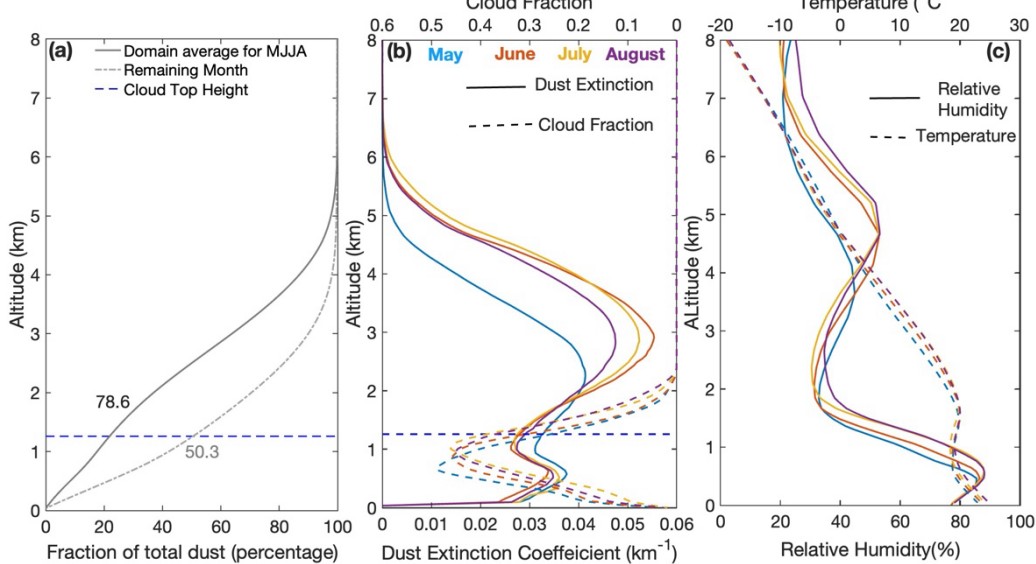

155

**Figure 2: (a) The domain mean profiles depict the fraction of dust concentration (in percentage) that lies below a particular altitude relative to the total column burden. Each point on the line represents the percentage of dust occurring below a specific altitude relative to the total column burden. The solid line represents the domain average profile for May to August, while the dashed line indicates the average values for the remaining months. The horizontal blue line denotes the mean low-level cloud top, and the numerical annotations beside each line indicate the dust percentage above the cloud top. (b) The domain-mean monthly profile of the dust extinction coefficient (solid lines) and low-level cloud fraction (dashed lines; note that the values go from right to left at the top of the figure). (c) The domain-mean monthly profiles of relative humidity (solid lines) and temperature (dashed lines). Colors in (b) & (c) indicate months with blue for May, red for June, orange for July, and purple for August. The datasets span from 2007 to 2017, with temperature and humidity profiles obtained from auxiliary meteorological data provided as part of CALIOP level 2 retrievals.**

We focused our analysis on the North Atlantic Ocean between May and August (2007-2017) when the
170 SAL dust frequently coexists with and typically above low-level cloud layers (Fig. 1b). Specifically, during this period, there is a higher probability of dust being present above the clouds than other periods of the year, allowing for enough data samples for statistically significant analysis (e.g., see supplementary Fig S1). When averaged over the domain of interest (see Fig. 1a), approximately 79% of the total column dust burdens are located above the mean low-level cloud top between May and August (Fig. 2a),
compared to only about 50% during other months. Compared to the May-August period, most of the above-cloud dust occurrences in other months are in March and April, with relatively lower overall dust optical depth (see red dots in Fig. 1b) and higher probability of mid-to-high level cloud occurrences that





can influence our analysis. Furthermore, the domain mean dust extinction profiles peak between May and August, with the column-integrated optical depth maximizing in June (~0.4; Fig. 2b and Fig. 1b). To this end, our analysis also examines the possible confounding influence of aerosol optical depth on the cloud response to dust-layer characteristics during the period.

## 2.1 Datasets

We obtained aerosols, clouds, and radiative fluxes information from satellite-based observations and meteorological parameters, including temperature and humidity, from reanalysis datasets. The satellite-based observations include those from CALIOP (Cloud-Aerosol Lidar with Orthogonal Polarization) onboard CALIPSO (Cloud-Aerosol Lidar and Infrared Pathfinder Satellite Observation) satellite, cloud profiling radar (CPR) aboard CloudSat, and reanalysis data from the European Centre for Medium-Range Weather Forecasts (ECMWF).

### 2.1.1 Aerosols and Cloud Data

We relied on CALIOP for vertically resolved measurements of clouds and aerosols (Winker et al., 2010). Equipped with an onboard lidar, this remote-sensing platform measures the backscatter signal of clouds and aerosols at two wavelengths, 532 nm and 1024 nm, and with distinct vertical resolutions: 30 meters from the surface to 8.2 km, 60 m from ~8.2 km to ~20.2 km, and 180 m above ~20.2 km. Cloud and aerosol features are identified using the contrast between backscatter signals from the target feature and the altitude-based threshold defined for the clear air (Vaughan et al., 2009). The backscattered signal at 532nm is preferred due to its sensitivity to a wider range of atmospheric targets, including small aerosol particles (Liu et al., 2004; Vaughan et al., 2009). The altitude boundaries of the identified features (aerosols or clouds) are determined iteratively using an algorithm known as the Selective Iterated Boundary Location Algorithm (SIBYL) (Hunt et al., 2009; Vaughan et al., 2002, 2009). The retrieved altitudes of the top and bottom boundaries of both cloud and aerosol layers have been extensively validated under different atmospheric conditions at various locations (Candlish et al., 2013; Lu et al., 2021a; Mamouri et al., 2009; Perrone et al., 2011). Compared to ground-based lidar measurement, aerosol, and cloud layer top boundaries are generally found within the range of 100 m (Kim et al., 2008; Lu et al., 2021b). However, several studies (Jethva et al., 2014; Rajapakshe et al., 2017) have observed that CALIOP underestimated the altitude of the bottom boundary of the aerosol layer compared to ground-based observations in earlier versions of the product. The latest release has improved issue (Lu et al., 2021).

We use the latest version 4 (V4), level 2 (L2) merged layer products of CALIOP, which have a 5-km horizontal resolution, to determine the dust geometric thickness and the vertical position of dust and cloud



layers (Liu et al., 2019). This version of CALIOP features an improved discrimination algorithm that can more effectively differentiate between aerosol and cloud compared to the previous version (Liu et al., 2009). In addition, the enhanced algorithm also has improved classification capabilities, particularly in detecting lofted dust and smoke layers (Liu et al., 2019), which were occasionally misclassified as cirrus

clouds in previous versions (Chen et al., 2010). Similarly, aerosol type discrimination has also undergone significant modification, using input variables such as location, surface type, estimated particulate depolarization ratio, and integrated attenuated backscatter to identify specific aerosol types (Kim et al., 2018a; Omar et al., 2009). Overall, the identified aerosol types in V4 products are marine, polluted continental/smoke, clean continental, polluted dust, elevated smoke, dusty marine, and pure dust (Kim et

al., 2018). Among these aerosol types, the identification of dust types (polluted dust, dusty marine, and pure dust) mainly depends on the estimated particle depolarization ratio, which quantifies the degree to which the light scattered by atmospheric particles undergoes depolarization. Specifically, layers with a high particulate depolarization ratio estimated to be greater than 0.20 are classified as pure dust, regardless of surface type, location, altitude, or integrated attenuated backscatter values. Conversely, moderately

depolarizing particles, with a depolarization ratio ranging from 0.075 to 0.20 and a layer base height below 2.5 km when observed over the ocean, fall into the category of dusty marine particles. When these identical, moderately depolarizing particles are detected above land or have a layer base height exceeding 2.5 km over ocean surfaces, they are labeled as polluted dust. Also, low depolarizing particles with a depolarization ratio of less than 0.075 are classified as polluted dust when found above desert surfaces

and have a 532nm integrated attenuated backscatter value greater than 0.0005. Because we focus on the SAL dust layer above the marine low-level clouds (generally above 2.5 km, see section 2.1.2 below), we used only aerosol layers identified as "pure dust" in this study. We consider only pure dust to avoid any contamination of sea salt aerosols in dusty marine aerosol types, which can be substantial, especially close to the ocean surface, and have a depolarization ratio like those used for dusty marine (Sakai et al., 2010).

We estimated the geometrical thickness and separation distance of the identified dust layer from the low-level cloud top using the 'Layer Top Altitude' and 'Layer Base Altitude' information provided in the L2 Merged Layer product (CAL_LID_L2_05km). Specifically, we estimated geometric thickness as the difference between the layer top and base altitudes of the dust layer, and similarly, we estimated the

240 separating distance as the difference between the dust-layer base altitude and the cloud-layer top altitude. Additionally, we obtained the dust optical depth as the aerosol optical depth (AOD) at 532nm for identified pure-dust-only profiles, with no other aerosol layers found above or below it at the same location. Although our estimated dust optical depth likely underestimates the column dust optical depth over the location since it considers only the pure dust layer, this underestimation has little to no impact

on this study's conclusion since our analysis focuses on the sensitivity of low-level clouds to a unit change in dust properties. For both aerosols and cloud parameters, we used only daytime retrievals for our analysis. We do so for two reasons: the first reason is that the daytime and nighttime retrievals for merged



CALIPSO-CloudSat products are limited only between 2006–2011 because CloudSat switched to daytime-only operation after October 2011 due to a battery malfunction (Nayak et al., 2012; Witkowski et al., 2018). Because our analysis is between 2007 and 2017, which extends past the October 2011 timeline of daytime-only products, we use only daytime datasets to maintain consistency across the observation period. The second reason is that our analysis focuses on understanding both the shortwave and longwave influences of dust-layer characteristics on clouds, in contrast to similar analyses of smoke aerosols (e.g., Herbert et al., 2020). Since nighttime analysis only provides the effect of longwave radiation, we use daytime data to assess the influence of both shortwave and longwave radiation by SAL dust on low-level clouds.

### 2.1.2 Radiative Flux Data

In addition to aerosol information from CALIPSO, we also obtained radiative fluxes and heating rates from merged CALIPSO-CloudSat datasets that leverage instruments on CALIPSO and CloudSat satellites. To estimate shortwave and longwave radiative effects, we used radiative fluxes and associated heating rates from the 2B-FLXHR-LIDAR product (Henderson et al., 2013; L'Ecuyer et al., 2008), which includes measurements from CPR aboard CloudSat, cloud and aerosol information from CALIOP, and other required variables from passive sensor MODIS aboard the Aqua satellite. For example, these radiative flux products used cloud parameters, including information on cloud ice and liquid water content, as well as effective radii, derived from CloudSat retrievals and the aerosol profile information obtained from CALIPSO. In addition, the radiative fluxes also use aerosol optical properties (such as asymmetry parameter and single-scattering albedo) obtained from D'Almeida et al. (1991), and surface properties, such as surface albedo and emissivity, are sourced from the International Geosphere-Biosphere Program (IGBP). Furthermore, atmospheric parameters (surface pressure, surface temperature, as well as profiles of pressure, temperature, specific humidity, and ozone mixing ratio) used for the radiative flux calculation were obtained from the European Centre for Medium-Range Weather Forecasts (ECMWF) reanalysis dataset and provided in the CloudSat dataset as ECMWF-AUX product (Cronk, 2017). The meteorological profiles and surface parameters from ECMWF 3-hour fields, given at a 0.5°×0.5° horizontal resolution, are interpolated to match the CPR geolocation, vertical bin, and time (Cronk, 2017). These aerosol and cloud parameters and the meteorological information are provided in a radiative transfer model called BugsRad (Fu and Liou, 1992) to obtain radiative fluxes at a vertical resolution of 240 m. The uncertainties of the obtained fluxes were assessed by performing multiple sensitivity analyses and comparing them with the top-of-atmosphere fluxes obtained from the Clouds and the Earth's Radiant Energy System (CERES) (Henderson et al., 2013). The biases, when compared to CERES fluxes, were determined by previous studies to fall within acceptable ranges (e.g., Henderson et al., 2013). The results of sensitivity studies point to CloudSat's estimates of liquid water content (LWC) being the primary source of error in shortwave fluxes, along with the assumed properties of undetected low clouds





contributing to a lesser extent (Henderson et al., 2013). On the other hand, for longwave fluxes, uncertainties predominantly stem from values assigned to skin temperature and lower tropospheric water vapor (Henderson et al., 2013). Because our study region is over the North Atlantic Ocean, the variabilities in skin temperature and lower-tropospheric water vapor are expected to be smaller and, therefore, result in smaller uncertainties in the overall radiative flux estimates than when compared to those over land.

We selected radiative flux products, including derived profiles of both upwelling and downwelling shortwave and longwave fluxes, along with corresponding heating profiles collocated with CALIPSO datasets for further analysis. These products comprise profiles of radiative fluxes, both with and without aerosols, under clear-sky and all-sky conditions. This enables us to compute the radiative effects of aerosols and clouds in both the shortwave and longwave spectra, which we then use to estimate the influence of dust-induced changes on fluxes and the heating rate at the cloud top. The dust-induced effect is assumed to be the aerosol-induced changes over locations with pure-dust-only profiles. We use the provided all-sky and all-sky no aerosol, upwelling, and downwelling fluxes to estimate the changes in radiative fluxes as follows:

$$\Delta F = \mathbf{F_{allsky}} - F_{allsky, no\ dust} \tag{1}$$

Where $\Delta F$ is the change in radiative fluxes due to the presence of dust, $F_{allsky}$ is net (downwelling minus upwelling) flux for all-sky conditions, and $F_{allsky, no\ dust}$ is the same, but computed with no dust (i.e., aerosol optical depth equals zero). This is estimated for both shortwave and longwave fluxes for each level given in the profile datasets. Subsequently, we calculate shortwave and longwave heating associated with the warming or cooling effect of dust as:

$$\frac{\partial \mathrm{T}}{\partial \mathrm{t}} = \left(\frac{1}{\rho C_p}\right) \cdot \left(\frac{\Delta F}{\Delta Z}\right)_{dust} \tag{2}$$

Where, $\frac{\partial T}{\partial t}$ is the heating rate (K day$^{-1}$), $\boldsymbol{\rho}$ =1.17 kg m$^{-3}$ density of atmospheric layer, $\boldsymbol{C_p}$ is the specific heat capacity of air at constant pressure (1004.67 J kg$^{-1}$ K$^{-1}$), and $\Delta \boldsymbol{Z}$ is the change in vertical levels (m). Details of how these heating rate estimates are used to understand radiative responses at the low-level cloud top, as well as used to benchmark the sensitivity analyses, are described in sections 2.2.3 and 2.2.4 below, respectively.

### 2.1.3 Meteorological data

To account for the influence of meteorological variability in our analysis, we used meteorological datasets, including temperature and humidity profiles, sea surface temperature (SST), and surface wind speed, derived from reanalysis products, such as ECMWF and MERRA-2. First, we used the CloudSat



ECMWF-AUX dataset, which is an intermediate product containing ECMWF atmospheric state variables and has been interpolated to match each CloudSat CPR profile and vertical bin (Cronk, 2017). We specifically used the temperature and pressure profiles from ECMWF-AUX to calculate potential temperature profiles and, subsequently, lower tropospheric stability (LTS)—an important predictor for low-level marine clouds (Wood and Bretherton, 2006). Second, to assess the large-scale influence on

cloud state, we obtained other key meteorological parameters, commonly known as cloud-controlling factors (CCF), such as SST and surface wind (10 m wind), from this dataset. These CCFs have been shown to influence low-level clouds over the North Atlantic region (Naud et al., 2023; Scott et al., 2020). Third, for consistency in the sensitivity analysis discussed in section 2.2.4, we also used the profiles of temperature, pressure, and relative humidity from vertically resolved meteorological information derived

from the Modern-Era Retrospective Analysis for Research and Applications, Version 2 (MERRA-2) (Gelaro et al., 2017), included in CALIOP datasets as input thermodynamic profile for simulation of radiative fluxes. The domain-averaged monthly mean profiles of temperature (dashed lines) and relative humidity (solid lines) are shown in Fig. 2c. While the temperature varies slightly, the humidity profiles show elevated relative humidity (about 50%) in the mid-troposphere that varies between May and August.

While such elevated moisture could have impacts on longwave (e.g.,Ryder, 2021), our methodology is designed to minimize such potential meteorological effects (see section 2.2.5), and further analysis suggests that it has minimal co-variability with dust-layer characteristics over our broad selected region, which is confirmed by our results (see section 3.4)

## 2.2 Analysis Procedure

We describe our methodology by first outlining the selection process for profiles with dust layers above clouds and categorizing dust-cloud configurations. Additionally, we detail the approach used to estimate cloud responses to dust layer properties and estimate how cloud and meteorological variabilities may influence our findings.

### 2.2.1 Identification of dust above clouds

We selected profiles that contained both dust and cloud layers, specifically focusing on cases where the dust layers were found above the low-level clouds. Here, we discuss the procedure used to identify such cases. First, we used feature classification flags in conjunction with the cloud layer-top and base altitude information from CALIPSO to identify locations that contained low-level clouds. For this, we focused on cases where the cloud layer-top altitude was less than 3.2 km (650 hPa) and the features were classified

as clouds. To avoid multiple cloud occurrences in the same location, including the influence of high-altitude clouds on low-level clouds (Christensen et al., 2013), we selected only locations with single-layer clouds from all other instances of low-level clouds. Second, we determine if these selected locations



contained any dust layers. We defined cases of dust above clouds when the base of the dust layer was above the cloud-top layer by at least 200 m. Following previous studies (Kim et al., 2008; Rajapakshe et
al., 2017), we used 200 m separation between the dust-base and cloud-top layers to account for the low confidence and potential uncertainties in the CALIOP retrieved aerosol base-layer height information. In addition, we excluded aerosol bases and tops extending beyond 6 km, as they are too far from the cloud top to have significant radiative effects (Herbert et al., 2020), and at these altitudes, cirrus clouds could also be misclassified as dust layers (Tackett et al., 2018).  Of the approximately 292,000 profiles with
single-layer low clouds examined over the 10 years (2007 to 2017) between May and August, about 178,000 of these profiles included dust layers. Among these 178,000 profiles, approximately 109,000 had dust layers residing above the cloud layer. The largest number of profiles having dust above the cloud is found in the eastern part of the selected domain (see Fig S2, which also shows when aerosols are found below the cloud layer and on both sides of the cloud layers for comparison). Because our analysis further
requires the spatiotemporal collocation with the radiative flux information from CloudSat, those requirements further limit the number of available profiles that can be used (see below).

## 2.2.2 Categorizing the dust-cloud relationship

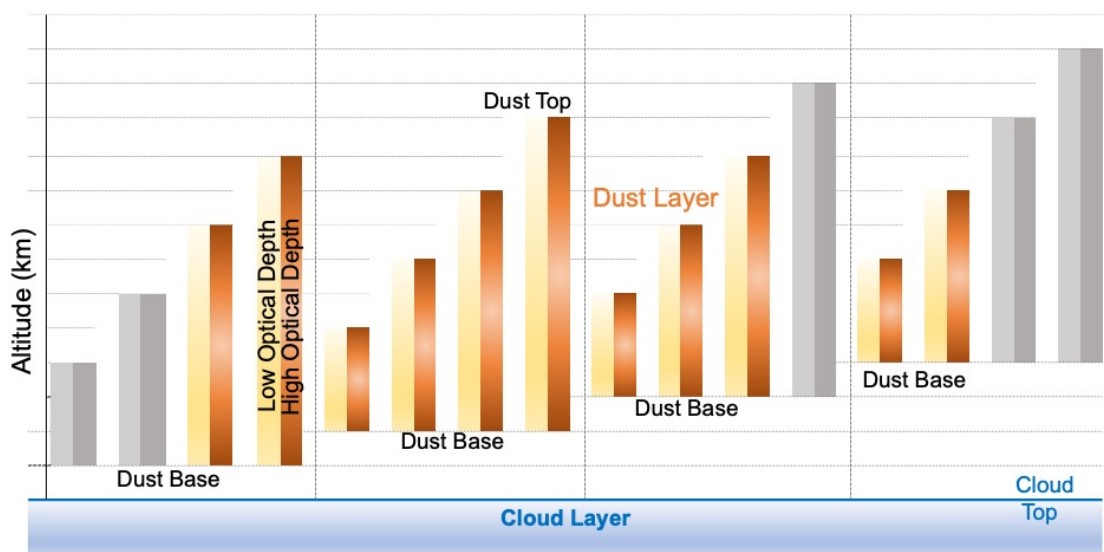

**Figure 3: Schematic showing our categorization of cloud, aerosol, and radiative heating**
**information, depending on the base and geometric thickness of the dust layer. This categorization includes dust base heights of 1, 2, 3, and 4km and geometrical thicknesses of 0.5, 1.5, 2.5, and 3.5 km, resulting in a total of 16 categories. Subsequently, each bar is split into two dust optical depth (DOD) regimes (low DOD between 0.05 and 0.2 and high DOD between 0.35 and 0.5). Gray bars do not meet the selection criteria discussed in section 2.3, thus resulting in 11 categories used for our**
**analysis.**



To understand the influence of the dust layer on the low-level cloud cover, we further divided our region of interest over the North Atlantic Ocean into 5°×5° grid boxes to minimize the effect of large-scale meteorological variability (Loeb and Schuster, 2008; Mauger and Norris, 2010; Scott et al., 2020). Previous studies have shown that meteorology may confound with aerosol-cloud interactions, even in cases where the aerosol layer is physically separated from the clouds (Arola et al., 2022; Loeb and Schuster, 2008; Mauger and Norris, 2010; Scott et al., 2020). One method for minimizing the meteorological influence on aerosol-cloud interactions is to limit the region of interactions, where it is assumed that meteorological variability is roughly invariant within the limited region (Adebiyi and Zuidema, 2018; Loeb and Schuster, 2008). We follow Loeb and Schuster (2008) and divide our region of interest into 5x5-degree grid boxes. While this approach may minimize meteorological variability confounding the aerosol-cloud interactions, it limits the availability of sufficient samples for each grid box. To account for this limited sampling while still ensuring a statistically significant analysis, we developed a framework for categorizing the dust-cloud relationship with in each 5°×5° grid boxes.

We categorized profiles having dust above the cloud layer in each 5°×5° grid box based on the dust layer base height and geometrical thickness only when aerosols, clouds, and radiative flux information are available. We stratified these profiles into four groups of dust base heights from 1, 2, 3, and 4 km, and for each of these base-height groups, we further divided them into four groups with geometrical thicknesses of 0.5, 1.5, 2.5, and 3.5 km. Therefore, within a 5°×5° grid box, a maximum of 16 categories can potentially exist, as illustrated in Fig. 3. To allow for robust estimates, only the 5°×5° grid boxes with a minimum of 30 profiles (no maximum limit) containing complete datasets of aerosol and cloud properties, meteorological parameters, and corresponding radiative fluxes, were retained for further analysis. As such, not all the 5x5 grid boxes contribute to each of the 16 potential dust-cloud categories in Fig. 3. For statistical significance and to determine which category to retain, we required at least six 5°×5° grid boxes over the North Atlantic Ocean to meet the minimum 30-profiles requirements to be averaged for each dust-cloud category. Therefore, the minimum number of profiles required for each dust-cloud category of Fig. 3 is at least 180 profiles, limiting the final number of categories to 11 distributed across the groups of dust base heights and geometric thicknesses (gold/golden brown bars in Fig. 3). These 11 dust-cloud categories still allow for the comparison of average cloud responses based on dust layers characteristics of geometrical thickness and base heights. Because dust layers with the same geometrical thickness or



base heights can exhibit varying dust optical depth, we further divided our categories into cases of low
and high optical depth with values between 0.05-0.2 and 0.35-0.5, respectively (see Fig. 3).

### 2.2.3 Estimating cloud and radiation responses to dust-layer characteristics

To understand the response of the low-level cloud cover to above-cloud dust characteristics, we first
computed grid-normalized cloud fraction obtained from CALIPSO and CloudSat separately for each of
the 11 categories within every participating 5°×5° grid box, using the following expression:

$$f' = \frac{\bar{f}_{cat}(x,y)}{\bar{f}_{lowcld}(x,y)} \tag{3}$$

Where $\bar{f}_{cat}(x,y)$ is the mean low-level cloud cover corresponding to each category at each grid box (x-
longitude and y-latitude), and $\bar{f}_{lowcld}(x,y)$ are the grid mean values averaged for the time period (May-
August, 2007-2017). We used normalized cloud fraction for each category because of instrumentation
differences in CloudSat and CALIPSO and because retrieval of these cloud variables is not done using a
similar approach, and as such their values are not consistent for the same longitude and latitude grid, as
similarly highlighted by previous studies (e.g., Bertrand et al., 2024; Kahn et al., 2008). For example,
CloudSat often underestimates low-level cloud cover due to its "surface clutter" zone, meaning it cannot
detect clouds very close to the ground, often common over the ocean, while CALIPSO's lidar can detect
thin, low-lying clouds that CloudSat might miss, leading to a higher low-level cloud fraction measurement
from CALIPSO (Bertrand et a. 2024). Therefore, normalizing the cloud fraction estimates allows us to
compare changes in cloud fraction to changes in dust layer characteristics (cloud fraction response) and
helps us understand whether such cloud sensitivity is robust across different datasets. In addition, although
not included in this study, we also analyzed other similar properties, such as cloud optical depth, for each
category, with similar conclusions to cloud fraction.

We used the variation in $f'$ across the categories to calculate cloud responses to changes in the dust-layer
base height, geometrical thickness, and dust optical depth. Specifically, we define estimating low-level
cloud response as the rate of change of grid-normalized cloud fraction per unit change in dust base height
($\frac{\partial f'}{\partial DB}\big|_{GT}$) and geometric thickness ($\frac{\partial f'}{\partial GT}\big|_{DB}$), when the other parameter is held constant. This can be
visualized as the difference in grid-normalized cloud fraction in Fig. 3 as the geometrical thickness
increases from left to right (for fixed base height) or as the dust-layer base height increases from left to
right (for the same geometric thickness). In addition, similar low-level cloud responses can also be
estimated for dust optical depth.

Similar to cloud fraction, we categorized the relevant radiative fluxes and associated all-sky and dust-
induced heating for each dust-cloud category. Specifically, we computed the radiative heating rate within
the dust layer and at the cloud tops, as well as all-sky radiative fluxes for shortwave and longwave





radiation at cloud tops for each category shown in Fig. 3. Similar to cloud response, we estimate the dust-induced cloud-top heating rate responses to a unit increase in dust base height $(\frac{\partial CldTopHt_{dust}}{\partial DB}\big|_{GT})$ and geometric thickness $(\frac{\partial CldTopHt_{dust}}{\partial GT}\big|_{DB})$.

### 2.2.4 Separating dust impacts from the cloud properties and underlying thermodynamic profiles

Although radiative heating estimates from CALIPSO–CloudSat products help establish the link between dust-induced cloud top heating and cloud response, the heating rate also depends on factors such as single-particle properties (e.g., extinction efficiency, single scattering albedo, asymmetry parameter) and the background temperature and humidity profiles. Moreover, dust-layer radiative heating is highly sensitive to absorption properties, such as particle size distribution and complex refractive index. For instance, dust layers dominated by coarse particles may exhibit stronger in-layer longwave cooling compared to those dominated by finer particles. In addition, distinct average thermodynamic profiles (of temperature and humidity) across categories (see Fig. S3) could also influence cloud-top radiative cooling and confound the observed dust-induced changes in cloud-top radiative cooling. Additionally, differences in cloud properties across categories may drive variations in cloud-top cooling independent of dust effects. Thus, to isolate the influence of dust properties, cloud properties, and thermodynamic profiles, we conducted sensitivity experiments using the Santa Barbara DISORT Atmospheric Radiative Transfer (SBDART) model(Ricchiazzi et al., 1998). SBDART is a widely used radiative transfer model that solves solar and thermal infrared radiation using the DISORT (Discrete Ordinates Radiative Transfer) algorithm. It allows the specification of surface types (e.g., ocean, desert, snow, forest) and user-defined atmospheric profiles for temperature, humidity, and pressure. The model also incorporates cloud parameters such as optical depth, cloud-top height, phase, and effective radius, enabling detailed simulation of radiative fluxes within and around cloud layers.

Using SBDART, we first recreate CloudSat-CALIOP radiative fluxes for each dust–cloud category in Fig. 3 using associated retrieved dust extinction profiles, cloud properties, and temperature and humidity profiles. The average atmospheric and dust/cloud properties for each category are shown in Fig. S3a & b and Table S1, including dust optical depth, cloud optical depth, and cloud top/bottom heights. To calculate single-particle optical properties (extinction coefficient, single scattering albedo, and asymmetry parameter), we applied Mie theory (Wiscombe, 1980), assuming spherical dust particles, using particle volume size distributions (Fig. S4a) and spectrally resolved complex refractive indices (Fig. S4b & c).

We considered three types of dust size distributions based on the measurements first collected by (Adebiyi and Kok, 2020) and recently standardized by Formenti and Di Biagio, 2024: first, a distribution typical of source regions with ~89% coarse-mode particles (>2.5 µm, Source - magenta line in Fig S4a); second,





a medium-range transported case with ~83% coarse-mode (MRT - red line in Fig S4a); and third a long-
465 range transported case dominated by fine particles (~50% coarse-mode, LRT; blue line in Fig S4a). To
capture variability in absorption due to dust aging, we used refractive indices from Di Biagio et al. (2019),
OPAC (Hess et al., 1998), and Volz, 1972, with shortwave imaginary refractive indices of 0.0026, 0.0075,
and 0.0080 at 0.5 µm wavelength, and longwave values of 0.3534, 0.4783, and 0.1620 at 10 µm
wavelength (Fig. S4b & c). These combinations yield nine cases with spectrally varying extinction
efficiency, single scattering albedo, and asymmetry parameter, which we used as inputs to SBDART to
examine how cloud-top heating responds to changes in dust size and absorptive properties (Fig. S4d, e &
f). We computed radiative fluxes for shortwave (0.25-4 µm), longwave (4-40 µm), and the net (0.25-40
µm) corresponding to each category based on average values of temperature and humidity profiles,
aerosols, and cloud parameters. From these calculated radiative flux profiles, we estimated dust-layer and
475 cloud-top heating/cooling rates, and the influence of dust on cloud-top heating/cooling rates.

To isolate the effects of dust properties, cloud properties, and thermodynamic profiles on cloud-top
radiative cooling, we performed 431,000 radiative transfer simulations using the SBDART model. These
simulations are performed for a combination of 11 thermodynamic (temperature and humidity) profiles,
11 dust structural categories (including dust-layer geometric thickness and base height), 11 cloud optical
depths, and 11 dust optical depths, across three spectral bands: shortwave (0.25–4 µm), longwave (4–
40 µm), and net (0.25–40 µm). We used 9 combinations of single-particle dust properties, resulting in
395,307 simulations. Additionally, we conducted 35,937 simulations with zero dust optical depth to
calculate dust-induced radiative heating profiles (as in equations 1 and 2 above). From these profiles, we
estimated all-sky and no-dust cloud-top cooling, as well as dust-induced heating at the cloud-top for all
485 three spectral bands. Since we used different dust-particle optical properties than those used in deriving
CALIPSO-CloudSat fluxes, we normalized the observed and simulated dust-induced cloud-top heating
by their respective mean across categories to enable meaningful comparisons. We further calculated the
sensitivity of dust-induced cloud-top heating to dust layer geometric thickness and base height using both
the normalized CloudSat-CALIPSO-derived heating and normalized SBDART-simulated heating.

To understand the dominating factors in dust-induced cloud-top heating, we quantified the sensitivity of
heating rates to dust layer geometric thickness and base height under three controlled experimental setups
(see Table S1). First, we isolated the effect of dust properties on cloud-top heating rates by allowing the
dust optical depth and dust structural category to vary across simulations, while holding the cloud optical
depth and thermodynamic profiles fixed. The resulting sensitivities reflect the influence of dust optical
properties alone, evaluated across 121 combinations of fixed values (11 cloud optical depth × 11
thermodynamic profiles; see experiment-2 in Table S1). Second, we isolated the effect of cloud properties
on cloud-top heating rates by fixing dust optical depth, thermodynamic profiles, and dust structural
category, and varying cloud optical depth and cloud top height, resulting in 1331 combinations (11 dust



optical depth × 11 dust categories × 11 thermodynamic profiles; see experiment-3 in Table S1). Third,
we assess the role of the thermodynamic profile by fixing dust optical depth, cloud optical depth, and dust
category, and allowing only the thermodynamic profiles to vary, again producing 1,331 combinations
(see experiment-4 in Table S1). As discussed earlier, to enable meaningful comparison across simulation
setups, we normalized all simulated dust-induced cloud-top heating values using the mean heating from
the full simulation set, where dust properties, cloud properties, and thermodynamic profiles were all
allowed to vary. We then used these three sets of normalized cloud-top heating values to estimate the
sensitivities of cloud-top heating rates to geometric thickness and dust base height, attributed specifically
to the effects of dust properties, cloud properties, or thermodynamic profiles, thereby enabling a robust
diagnosis of their contributions to dust-induced cloud-top heating rates.

**2.2.5 Quantifying the role of dust particle size and absorption**

To examine how particle size distribution and refractive index influence dust-induced cloud-top heating,
we calculated all-sky, no-dust, and dust-induced heating for all nine combinations (Fig. S4) of dust single-
particle optical properties, derived from three representative size distributions, source-region dominated
(89% coarse-mode), medium-range transported (83% coarse-mode), and long-range transported (50%
coarse-mode), and three sets of spectrally varying complex refractive indices (from Di Biagio et al., 2019;
OPAC, Hess et al., 1998; and Volz, 1972). For each combination, we computed the sensitivity of dust-
induced cloud-top heating to dust optical depth under fixed cloud optical depth, cloud-top height, and
thermodynamic profiles. These simulations quantify how different dust compositions, characterized by
varying absorption and scattering efficiencies, modulate the radiative impact of dust layers on low-level
clouds. The resulting nine dust optical depth sensitivities help estimate the effect of size and dust-
longwave absorptivity.

**2.2.6 Estimating the influence of meteorology**

In tropical and subtropical oceanic regions, meteorological factors commonly known as cloud-controlling
factors, such as sea surface temperature (SST), inversion strength, subsidence, and surface wind speed,
influence the cloud cover  (Klein et al., 2017; Scott et al., 2020).  For example, a higher SST typically
leads to an expansion of the boundary layer heights, reducing cloud formation and persistence, while
stronger inversion strength limits dry air entrainment, promoting larger cloud cover. Subsidence often
quantified using vertical velocity at 700 hPa or adjacent layers above the cloud (Adebiyi et al., 2015;
Naud et al., 2023), generally correlates positively with cloud fraction. However, when inversion strength
is constant or absent, stronger subsidence may reduce cloud cover (Adebiyi and Zuidema 2018).
Similarly, surface wind speed, a frictional driver of evaporation, theoretically should lead to an increase
in cloud fraction with enhanced wind speed. Nevertheless, wind speed is also tied to temperature and





moisture advection, resulting in an unclear relationship with cloud cover. Thus, it becomes necessary to analyze these cloud-controlling factors to establish the causal effect of dust on low-level clouds. As highlighted above, to minimize the effect of such cloud-controlling factors in our analysis we restricted our analysis to small grid boxes (5°×5° grid box), where we assume meteorological variability is largely invariant and considered the same season.

Further, to understand the effect of these confounding meteorological variables, we compared the anomalies of these cloud-controlling factors such as surface wind speed, sea-surface temperature, inversion strength, and humidity, across the categories in Fig. 3 (see Fig. S5 for respective spatial patterns). A negligible and non-systematic variation in these parameters across the categories would suggest their minimal influence on estimated low-level cloud responses. For example, the radiative response of above-cloud humidity, which can affect the cloud by entraining into the cloud layer and can increase downwelling longwave radiation reaching the cloud top, may resemble the longwave effect of dust and could be difficult to separate from the radiative effect of above-cloud dust. We adopted an approach discussed by Eastman and Wood (2018) to estimate the entraining humidity, defined as specific humidity in the layer directly above the inversion top, and the radiating humidity, defined as the mean humidity above the inversion top and below 700 hPa. The inversion top in the present study was considered a layer just above the cloud top instead of being estimated from the potential temperature profile.

## 3 Results

We found that low-level clouds are sensitive to the dust-layer altitude, geometric thickness, and dust optical depth over the North Atlantic, and this sensitivity is associated with the longwave radiation induced by the above-cloud dust layer. The details of these results are presented below, including in section 3.1, where we discuss the cloud response to changes in dust base (DB) and geometric thickness (GT); section 3.2, where we isolate the influence of dust optical depth (DOD) on these cloud responses for each combination of DB and GT, and section 3.3 where we estimate the contributions of the DB, GT, and DOD to low-level cloud response. In section 3.4, we explain how the perturbations in these dust characteristics (DB, GT, and DOD) affect cloud-top radiative fluxes and, consequently, modify cloud-top longwave cooling that facilitates the observed changes in the low-level cloud cover. Finally, we examine in section 3.4 the role of meteorological variability that may affect the instantaneous radiative heating rates and the large-scale environment that may confound with the influence of dust characteristics on the low-level cloud response.

## 3.1 Low-level cloud response to dust layer characteristics.




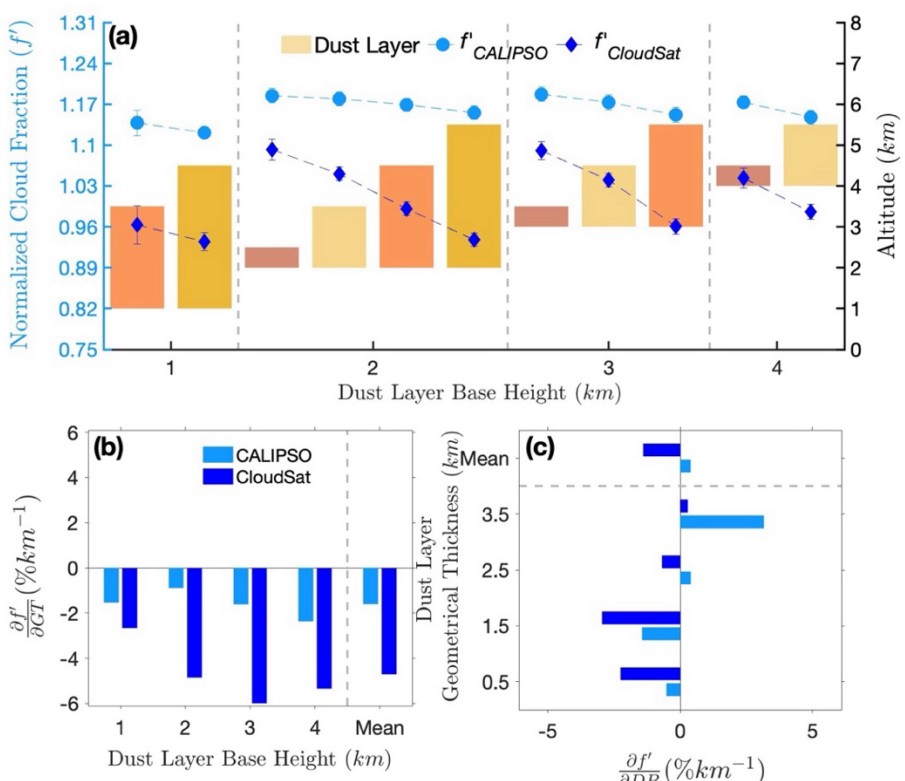

**Figure 4: Variation and response of low-level grid-normalized cloud fraction to the geometrical thickness and dust layer base. (a) The mean grid-normalized cloud fractions ($f'$) for CALIPSO and CloudSat (circular and diamond markers, respectively) for each category defined in Figure 3 (see section 2.2.3 for details). The bars represent categories with the same base height group, and those with the same geometric thickness are represented by the same colors across the different groups. The height of the bars corresponds to geometrical thickness: 0.5 km (violet), 1.5 km (light yellow), 2.5 km (orange), and 3.5 km (yellow) at the dust-layer base of 1, 2, 3, and 4 km. Error bars indicate the standard error. (b) Low-level cloud response to dust-layer geometrical thickness $\left(\frac{\partial f'}{\partial GT}; \% \, km^{-1}\right)$, shown for different dust-layer base heights. (c) Low-level cloud response to dust base height $\left(\frac{\partial f'}{\partial DB}; \% \, km^{-1}\right)$ for different geometrical thicknesses. Light blue bars represent the low-level cloud response for CALIPSO, and dark blue bars are for CloudSat.**

As shown by previous studies (e.g., DeFlorio et al., 2014; Doherty and Evan, 2014) and highlighted earlier in Fig. 1d, low-level cloudiness typically increases due to dust's semi-direct effect over the North Atlantic Ocean when the dust is above the cloud layer. However, here, we found that the extent of this increase also depends on the dust layer's relative altitude and geometrical thickness. To understand this relationship, we categorized the low-level cloud fraction based on dust base (DB) and geometric thickness (GT), and the resulting categories are shown in Fig. 4a for grid-normalized cloud fraction ($f'$) – the cloud



fraction for each category normalized by the climatology over each grid box – for CloudSat and CALIPSO (see section 2.2.3 above). The result shows that the domain-mean grid-normalized cloud fraction ($f'$) is greater than unity for CALIPSO and CloudSat (1.16 (±0.12) and 1.01 (±0.27), respectively). As highlighted earlier (see section 2.2.3), the difference between estimate of cloud fraction in CloudSat and CALIPSO is likely associated with differences in instrumentation (radar versus lidar), resulting in different limitations and uncertainties in the retrieved cloud fraction, including underestimation of CloudSat low-level cloud cover due to its difficulty to detect clouds very close to the ocean surface (e.g., Bertrand et a. 2024).

Despite this difference in retrieved cloudiness, the cloud cover decreases as a function of dust-layer geometric thickness, regardless of the dust-base altitude, in both datasets. Specifically, when the dust-base height is 1 km, for an increase of geometric thickness from 2.5 km to 3.5 km, the reduction in grid-normalized cloud fraction ranges by about 1% (1.13 to 1.12) for CALIPSO and about 3 % (0.96 to 0.93) for CloudSat. Similarly, when the dust-base height of 2 km and dust-layer geometric thickness increases between 0.5 and 3.5 km, the reduction in grid-normalized cloud fraction ranges by about 3% (from 1.18 to 1.15) for CALIPSO and about 16% (1.09 to 0.93) for CloudSat (see Fig 4a). When the dust-base height is 3 km (geometric thicknesses between 0.5 and 2.5 km) and at 4 km (geometric thicknesses between 0.5 and 1.5 km), the reduction in the grid-normalized cloud cover ranges by about 3% and 3% for CALIPSO and 13% and 6% for CloudSat, respectively. Overall, all categories of dust base height and geometric thickness show a reduction in grid-normalized cloud fractions, but the ranges of the reduction for CloudSat are consistently larger than those for CALIPSO, indicating the much larger variability in CloudSat low-level cloud fraction over the North Atlantic Ocean. Although the magnitudes of these changes in cloudiness are small, they are statistically significant and indicate that changes in dust-layer geometric thickness influence low-level cloudiness.

Because the classes of dust-layer geometric thickness are not the same for the different dust-base altitudes (see Fig. 3 & Fig. 4a), we further isolate the cloud response to changes in dust-base height separate from the geometric thickness by examining how the grid-normalized cloud fraction changes per unit change in one parameter while holding the other relatively constant (Fig. 4b & c). The result shows that a unit increase in dust-layer geometric thickness consistently leads to a reduction in cloudiness regardless of the dust-base height (Fig. 4b). On average, cloud fraction decreases by 1.7% and 5.1% for each kilometer increase in dust-layer geometric thickness based on CALIPSO and CloudSat, respectively (see the last bars in Fig. 4b). Additionally, the low-level cloud response to unit change in geometric thickness is stronger for higher dust-base altitudes. Specifically, for each kilometer increase in geometric thickness, the low-level cloud response is -1.54%, -0.89%, -1.16%, and -2.37% for dust layers with base heights of



1 km, 2 km, 3 km, and 4 km, respectively, for CALIPSO and are -2.66%, -4.86%, -5.99%, and -5.34% for CloudSat.

In addition, the grid-normalized low-level cloud response to one unit change in dust-base height depends on the dust-layer geometric thickness. For relatively thinner dust layers (GT ≤ 1.5 km), we find that an increase in dust-base height tends to decrease cloud fraction. Conversely, for relatively thicker dust layers
(GT ≥ 2.5 km), we find that an increase in dust-base height results in an increase in cloud fraction (see Fig. 4c). Although this inference is generally consistent with varying magnitude across the datasets, for a geometric thickness of 2.5 km, the cloud response to changes in dust-base height is negative for CloudSat. Specifically, the low-level cloud response is positive (0.40%/km) in CALIPSO but negative (-0.69%/km) in CloudSat. In addition, for dust-layer geometric thickness greater than ~3.5, the cloud response in
CALIPSO is substantially larger than that of CloudSat, although of the same sign. These inconsistencies may be attributed to the differences in retrieved cloud fraction between CALIPSO and CloudSat resulting in different variabilities that are exacerbated by the selection procedure (see e.g., Fig. S6). Overall, for thinner dust layers, one unit increase in dust-layer base height reduces cloud cover by 0.52% and 1.44% for geometric thickness values of 0.5 km and 1.5 km, respectively, using CALIPSO data and 2.25% and
2.95%, respectively, using CloudSat. In contrast, for thick dust layers (GT=3.5 km), increasing the base height leads to an increase in cloud response, with values of 3.17% and 0.30% from CALIPSO and CloudSat data, respectively.

**3. 2 Influence of dust optical depth in cloud response to dust layer characteristics.**



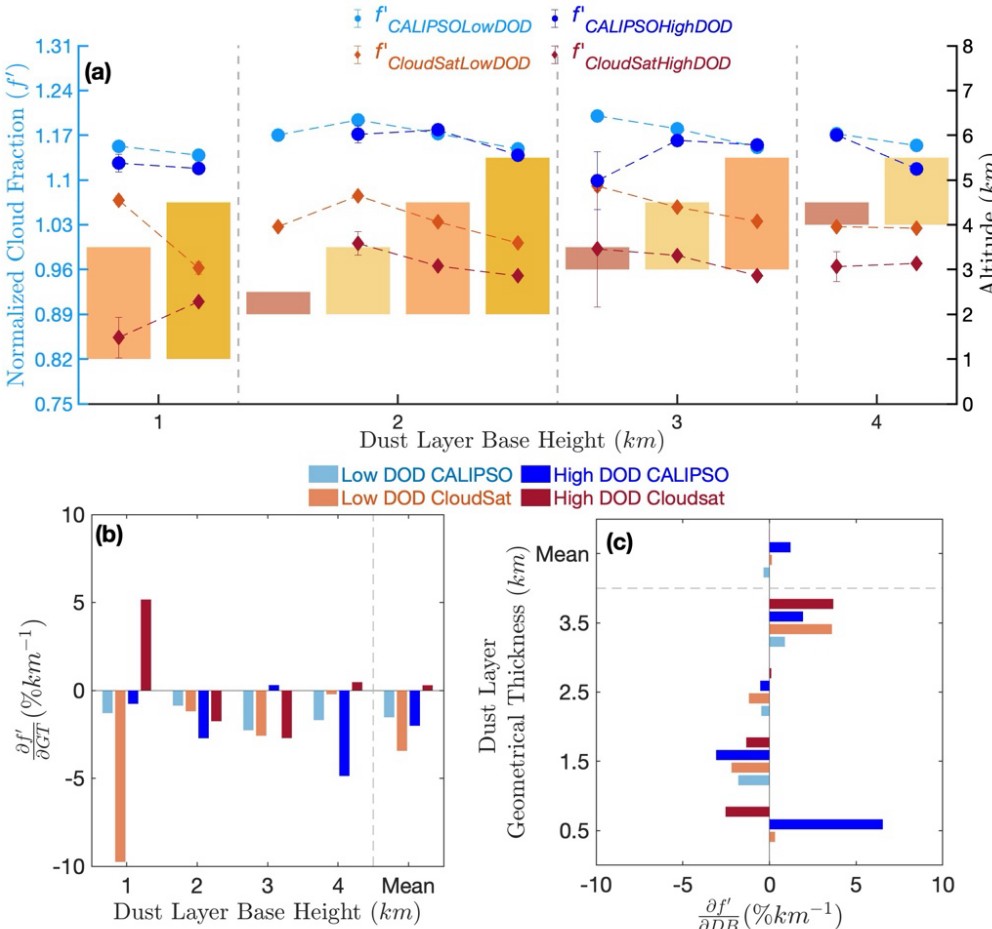

**Figure 5: (a) The mean grid-normalized cloud fraction (f') separated into low (0.05-0.2; μ±σ = 0.12 ± 0.02; light blue and light red lines) and high (0.35-0.5; μ±σ = 0.41 ± 0.02; dark blue and dark red lines) dust optical depth (DOD) for the categories defined in Fig 3. Circular markers with blue colors are for CALIPSO, and diamond markers with red colors are for CloudSat. The bars represent categories with the same base height group, and those with the same geometric thickness are represented by the same colors across the different groups. The height of the bars corresponds to geometrical thickness: 0.5 km (violet), 1.5 km (light yellow), 2.5 km (orange), and 3.5 km (yellow) at the dust-layer base of 1, 2, 3, and 4 km. Error bars indicate the standard error. (b & c) Low-level cloud response to (b) dust layer geometrical thickness and (c) dust base heights separated for low and high DOD cases. Blue bars are for CALIPSO, and red bars are for CloudSat. Light colors show responses corresponding to low DOD values, while dark colors are for high DOD.**

Because dust-base altitude and geometric thickness may also co-vary with dust optical depth (DOD), we further divided our categories (shown in Fig. 4a) into subsets based on profiles with low (0.05–0.2; μ±σ



= 0.12 ± 0.02) and high (0.35–0.5; μ±σ = 0.41 ± 0.02) DOD values (see Fig S7) and estimated the
corresponding grid-normalized cloud fraction (see Fig. 5a). We find that the variations in cloud fraction
as a function of dust-base height and geometric thickness for both low and high DOD cases broadly
remain consistent with the patterns observed for all dust (compare Fig. 5a and Fig. 4a). Specifically, the
cloud fraction generally decreases as geometric thickness increases. Additionally, the grid-normalized
cloud fractions for the same category are usually lower for high DOD cases than for low DOD cases. This
suggests that cloud fraction responds negatively to increases in DOD, contrasting with the cloud response
to biomass-burning aerosols that show an increase with higher aerosol optical depth (Adebiyi and
Zuidema, 2018). In other words, while cloud response to DOD typically results in enhancement of low-
level cloudiness (i.e., cloud response to above-cloud dust semi-direct effect; as shown in Fig 1d and Fig.
4a), that enhancement decreases as DOD increases, with stronger enhancement occurring for low DOD
than high DOD cases. When averaged across all categories, the mean grid-normalized cloud fraction
differs between low and high DOD cases by about 3% (1.17 ± 0.02 and 1.14 ± 0.02) for CALIPSO and
about 9% (1.04 ± 0.04 and 0.95 ± 0.04), respectively.

Furthermore, we also find that the cloud fraction decreases with increasing DOD and is broadly consistent
regardless of the range used to define low and high DOD cases. Specifically, we find negative low-level
cloud changes to increasing DOD are consistent across all combinations of dust-base height and geometric
thickness (e.g., Fig. S8). Consequently, we estimate cloud response to DOD as the change in grid-
normalized low-level cloud fraction per unit change in DOD (Fig. S8). The result shows that the mean
low-level cloud fraction changes by approximately -17 % per unit change in DOD. Similarly, the cloud
response to DOD ranges from -81% to +2%, depending on the combination of dust-layer base altitude
and its geometric thickness (see Fig. S8). While low-level cloud cover is sensitive to increases in DOD,
the results suggest that potential co-variabilities between dust-layer characteristics and DOD do not
significantly influence the cloud response to dust-base height and geometric thickness.

Therefore, we separate the influence of dust-base height and geometric thickness on low-level clouds for
low and high DOD cases (Fig. 5b & c; as done above in Fig. 4b & c). Our result shows that the low-level
cloud response to dust-layer geometric thickness ($\frac{\partial f'}{\partial GT}$) is stronger for high DOD cases, despite having
lower grid-normalized cloud fraction, than for low DOD cases (Fig. 5b). The mean cloud responses to
geometric thickness are -1.53%/km and -2.01%/km of the mean $f'$ for low and high DOD cases,
respectively, for CALIPSO. In contrast, for CloudSat, the mean cloud response to geometric thickness is
stronger for low-DOD cases (-3.44%) compared to high-DOD cases (0.29%). However, this difference
in CloudSat, when compared with CALIPSO, is because of the large variation in cloud response for the
dust layer at 1 km base (see first column in Fig. 5b). Removing that extreme case, our results show
consistency between CloudSat and CALIPSO and indicate that the low-level cloud response to geometric
thickness remains negative for low and high DOD cases. Unlike the low-level cloud response to





geometric thickness, the mean cloud response to dust-base height ($\frac{\partial f'}{\partial DB}$) is relatively small due to

contrasting signs for low and high DOD cases (Fig. 5c). For CALIPSO, the mean cloud responses are -0.34% for low DOD and 0.14% for high DOD, and, for CloudSat, the mean responses are 1.22% for low DOD and -0.02% for high DOD.

## 3.3 Contributions of dust-layer characteristics and optical properties to low-level cloud response

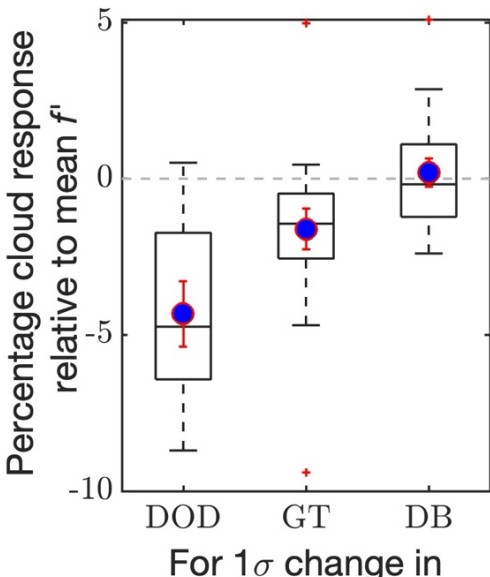


**Figure 6: Box-and-whisker plot showing the percentage change in cloud response relative to the mean f' for a 1σ (1 standard deviation) change in dust optical depth (DOD; 1σ = 0.27), geometric thickness (GT; 1σ = 0.964 km), and dust base height (DB; 1σ = 0.778 km). Blue markers represent the mean values, with error bars indicating the standard error (red).**

Because of the different units in the cloud response to dust-layer characteristics (base height, DB, and geometric thickness, GT) and dust optical depth (DOD), it is difficult to assess their relative contributions to the low-level cloud responses. To understand the relative contribution of dust-layer GT, DB, and DOD, we estimate the sensitivity in the low-level cloud response to a one-standard-deviation (1σ) increase in each parameter relative to the basin-wide mean cloud fraction ($\sigma_X \frac{\partial f'}{\partial X}$; where X is DOD, GT or DB and

$\sigma_x$ is their standard deviations). As shown above, and for one unit change in each parameter (DOD, GT, and DB), the low-level cloud responds negatively with the strongest cloud response to the unit change in DOD (see Fig. 6). Specifically, we find that relative reduction in cloud response to 1σ change in DOD is about -4.3 (± 1.04) %, compared to -1.6 (± 0.65) % and 0.19 (± 0.45) % to 1σ change in dust-layer geometric thickness and dust base respectively. In other words, if all parameters remain constant, 1σ



change in DOD will reduce the low-level cloud fraction by about a factor of 3 more than 1σ change in GT, and significantly more than 1σ change in DB that is largely negligible within the error range. Overall, our results, examining the effects of dust-layer dust-base heights, geometric thicknesses, and dust optical depth on low-level cloud cover, indicate that low-level clouds respond negatively to the dust properties and characteristics, but more strongly to increases in dust optical depth and geometric thickness, than to

the same increases in dust base height.

## 3.4 Radiative influence of dust-layer characteristics and optical properties on low-level cloudiness

We hypothesize that the variations in radiative fluxes at the cloud top drive the observed cloud responses to perturbations in dust-base heights, geometric thickness, and dust optical depth. This hypothesis builds on the traditional mechanism of aerosol semi-direct effect, whereby changes in cloud response to above-

cloud absorbing aerosols occur through radiative influence at cloud-top, such as changes in cloud-top entrainment (Wilcox, 2010). As highlighted in the introduction (section 1), the mechanisms of dust semi-direct effect are expected to deviate from traditional mechanisms associated with other absorbing aerosols, such as black carbon, because the pathways of dust-radiation interactions now include the longwave radiation that is generally absent in other absorbing aerosols. Within this context, therefore, we

seek to understand the radiative influence of dust-layer characteristics on low-level cloudiness. To do so, we examined how changes in dust-base heights, geometric thickness, and dust optical depth influence radiative heating rates within the dust layer and at the underlying low-level cloud top, and consequently, the relationship with the low-level cloudiness.

### 3.4.1 Changes in dust-induced radiative impacts due to dust layer characteristics



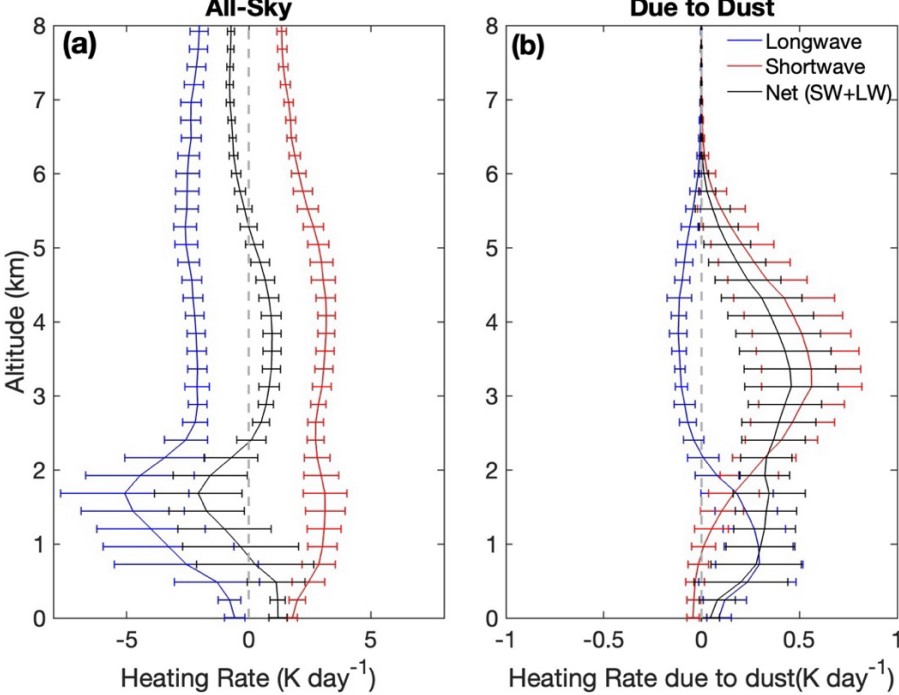

**Figure 7: The domain-mean heating rate profiles over the North Atlantic Ocean for (a) all-sky and (b) due to dust, calculated from the difference between all-sky with and without dust radiative fluxes (see section 2.1.2). Red lines are shortwave (SW) heating rates, blue lines are longwave (LW) heating rates, and black is net (SW+LW). Error bars indicate the standard deviation for each level.**

Using the CALIPSO-CloudSat heating rates profile averaged over the North Atlantic Ocean (see box in Fig. 1), Fig. 7 shows that the dust layer above the low-level cloud is well-defined between approximately 2.5 and 6 km with dust-induced net radiative warming at the low-level cloud top. Within the dust layer, the all-sky shortwave warming dominates the longwave cooling in the net heating rates (Fig. 7a). In addition, these all-sky heating rates are primarily because of the dust-radiation interaction within the layer, which shows levels of maximum longwave emission slightly above levels of maximum shortwave extinction (Fig. 7b). Specifically, the peak dust-induced shortwave warming of 0.56 (±0.26) K day$^{-1}$ occurs at approximately 3.12 km, whereas dust-induced longwave cooling peaks at approximately 3.82 km with a value of -0.12 (±0.04) K day$^{-1}$ (Fig. 7b).

In contrast to the dust layer, the all-sky net heating rates within the boundary layer are dominated by longwave cooling, and slightly opposed by shortwave warming, close to the cloud top as expected for low-level clouds (Fig. 7a). However, we find that the presence of dust layer above the cloud induces an anomalous longwave warming reinforcing the all-sky shortwave warming and counteracting the mean longwave cooling within the boundary layer. In other words, dust-induced longwave warming effectively




weakens the net cloud-top radiative cooling when dust is present above the low-level clouds than when it is not. In addition, this anomalous dust-induced longwave warming peaks at approximately 0.723 km with a value of 0.30 ±0.22 K day⁻¹ (Fig. 7b). Overall, the profiles show that dust-induced shortwave warming dominates net heating rates within the dust layer and dust-induced longwave warming weakens the dominant mean longwave cooling within the boundary layer. It is also worth noting that the uncertainties in the net heating rates are dominated by the uncertainties in shortwave heating rates within the dust layer and the uncertainties in the dust-induced longwave heating rates within the boundary layer (see error bars in Fig. 7).

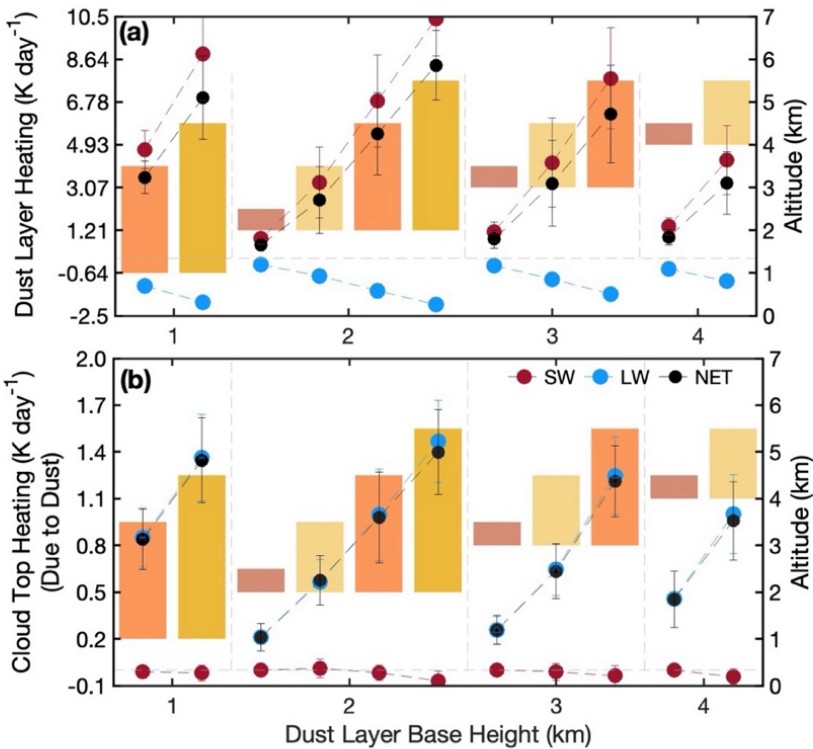

**Figure 8: Dust-induced radiative heating (a) within the dust layer and (b) at the low-level cloud top, separated into categories based on dust-layer base height and geometric thickness. Dashed lines are the radiative heating rates of shortwave (SW; red lines), longwave (LW; blue lines), and net (SW+LW; black lines). The bars represent categories with the same base height group, and those with the same geometric thickness are represented by the same colors across the different groups. The height of the bars corresponds to geometrical thickness: 0.5 km (violet), 1.5 km (light yellow), 2.5 km (orange), and 3.5 km (yellow) at the dust-layer base of 1, 2, 3, and 4 km. Error bars indicate the standard error.**

Separating the heating rates as a function of the dust-layer characteristics, we find that changes in the dust-induced heating or cooling rates within the dust layer and within the low-level clouds vary as a





function of the geometric thickness of the dust layer (Fig. 8 and Fig. S9). Specifically, our results indicate that dust-induced shortwave warming continues to dominate over longwave cooling within the dust layer, with both strengthening with increasing geometric thickness of the dust layer (Fig 8a). Between the dust-layer categories, the mean all-sky shortwave heating within the dust layer as a function of geometric thickness ranges up to about 10.3 K day$^{-1}$, partially offset by the mean longwave cooling that can also range up to 2.01 K day$^{-1}$ (e.g., for a dust base height of 2 km in Fig. 8a). In addition, for each category, the shortwave heating peaks around the middle of the dust layer whereas the longwave cooling peaks slightly above it (Fig S9), similar to the basin-wide mean profiles in Fig. 7. In contrast to the dust layer, dust-induced longwave warming within the low-level cloud dominates over the shortwave cooling, and the heating rates also increases with the dust-layer geometric thickness (Fig 8b & Fig S9). For example, the increase in heating rates ranges between ∼ 0.2 and 1.46 K day$^{-1}$ for dust-base height at 2 km as the geometric thickness increases from 0.5 to 3.5 km.

Similar to the heating rate changes as a function of geometric thickness for the same dust-base heights, there are also notable changes in the dust-induced heating rates as a function of dust-base altitudes for the same geometric thickness. For example, with dust layer geometric thickness of 2.5 km, the dust-induced longwave warming at the cloud top increases from around 0.85 K day$^{-1}$ to around 1.25 K day$^{-1}$ between the dust base of 1 to 3 km (see orange bars in Fig. 8b). As indicated above and regardless of the dust-layer configurations, these increases in dust-induced longwave warming reduce the mean longwave cooling near the cloud top (compare Fig. 8b and Fig. 7), with a more significant reduction for thicker dust layers. Overall, our results indicate that dust-induced longwave warming accounts for an average decrease of -9.3 % (up to -18.7 % depending on dust-layer configuration) of the overall net radiative cooling near the low-level cloud top over the North Atlantic Ocean.

### 3.4.2 Influence of dust optical depth on dust-induced heating rates




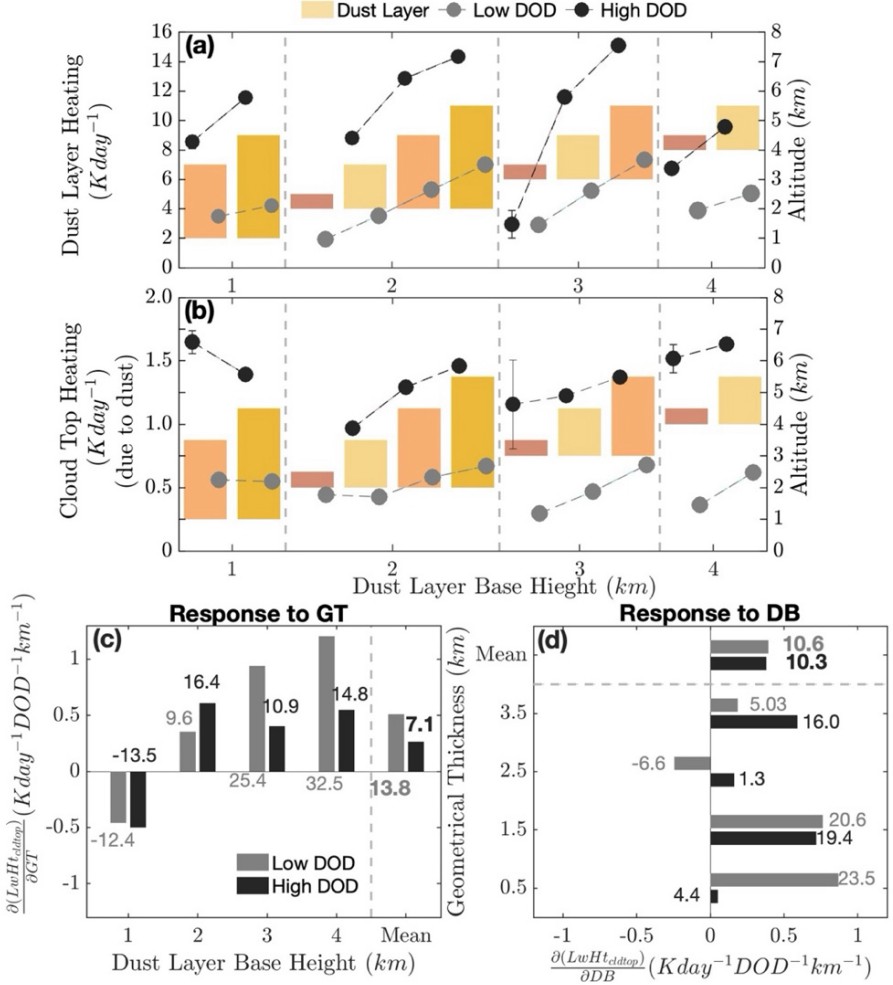

**Figure 9: The net (shortwave + longwave) heating rate (K day⁻¹) within (a) the dust layer and at (b)**
**the cloud top are categorized by dust layer base height and geometric thickness. Grey circles**
**indicate heating rates for low dust optical depth, while black circles indicate high dust optical depth.**
**Error bars represent the standard error. The bars represent categories with the same base height**
**group, and those with the same geometric thickness are represented by the same colors across the**
**different groups. The height of the bars corresponds to geometrical thickness: 0.5 km (violet), 1.5**
**km (light yellow), 2.5 km (orange), and 3.5 km (yellow) at the dust-layer base of 1, 2, 3, and 4 km.**
**Dust-induced cloud-top heating response to (c) dust layer geometrical thickness**
$\left(\frac{\partial LwHt_{cldtop}}{\partial GT}; K\,day^{-1}DOD^{-1}km^{-1}\right)$**, shown for categories with differing dust-layer base heights,**
**and (d) dust base height** $\left(\frac{\partial LwHt_{cldtop}}{\partial DB}; K\,day^{-1}DOD^{-1}km^{-1}\right)$**, displayed for categories with**
**differing geometrical thicknesses. Blue bars are for low DOD, and red bars are for high DOD.**
**Numbers annotated with bars are percentages relative to the mean value of dust-induced cloud top**
**heating (3.7±0.7 K day⁻¹ DOD⁻¹). Mean percentage values are written in bold font.**



Although the above result shows changes in radiative heating rates due to dust-layer geometric thickness for different dust-base heights, the dust-induced heating/cooling within the dust layer and at the cloud top also varies with dust optical depth. Similar to Fig. 5 above, we separated the radiative heating rates into cases of low and high DOD (Fig. 9 and Fig. S10). Within the dust layer, the all-sky net warming is significantly higher for high DOD than for low DOD cases. For low DOD, the net warming in the dust layer is 4.5 ±1.6 K day$^{-1}$, with 20.3 K day$^{-1}$ from shortwave warming and -15.8 K day$^{-1}$ from longwave cooling (see Fig. 9a & S11). This net warming with the dust layer doubles (to about 9.8 ±3.8 K day$^{-1}$) for high DOD cases, with 29.7 K day$^{-1}$ from the shortwave warming and -19.9 K day$^{-1}$ from longwave cooling (Fig. 9a & S11). Essentially, shortwave warming increases faster per unit increase in dust optical depth than longwave cooling (Fig. S11). Despite the significant change in dust-layer shortwave warming, the dust-induced radiative heating changes near the cloud top are controlled primarily by the longwave component, both for low and high DOD cases (Fig. 9 & S11). Near the cloud top, the overall all-sky longwave cooling does not significantly differ (Fig. S11), but the dust-induced longwave warming is more than doubled between low and high DOD cases (Fig. 9b). Specifically, the average dust-induced cloud-top heating rates, dominated by dust-induced longwave warming, show values of 0.51 (±0.12) K day$^{-1}$ for low DOD case and 1.34 (±0.22) K day$^{-1}$ for high DOD case (Fig. 9b). Our result of dust-induced warming near cloud top is consistent regardless of DOD classification or range used (Fig. S12). As a result, we calculated the heating rate response of the dust-induced cloud-top radiative warming to a unit change in DOD ($\frac{\partial LwHt_{cldtop}}{\partial DOD}$), and found a consistent pattern of dust-induced warming for all combinations of dust-base height and geometrical thickness.

Regardless of DOD, dust-induced longwave warming at the cloud-top remains dependent on base height and the geometric thickness of the dust layer, as highlighted by Figure 9. To further understand the dependence of cloud-top longwave radiative heating changes on the dust-layer characteristics, we estimated the dust-induced longwave warming response to dust base height and the geometric thickness for low and high DOD cases, similar to the low-level cloud response (compare Fig. 9c&d to Fig. 5b & c). We find that the dust-induced DOD-normalized cloud-top longwave warming response per unit increases in geometric thickness ($\frac{\partial LwHt_{cldtop}}{\partial GT}$) is positive for most dust base heights (Fig. 9c). In other words, for every increase in dust geometric thickness, the dust-induced longwave warming increases per unit DOD at the cloud top. The only exception is for the closest dust layer to the cloud top, where there is a higher possibility of longwave cooling within the dust layer extending to the cloud top as the geometric thickness increases and also there are higher uncertainties in estimated low-level cloud responses (cf. Fig. 5, S6, and Fig. S12). Overall, the dust-induced cloud-top longwave warming response per unit DOD to increase geometric thickness is higher for low DOD cases, which is 13.8% of the mean dust-induced cloud top heating (3.7±0.7 K day$^{-1}$ DOD$^{-1}$), than high DOD cases, which amounts to 7.1%. Similar to the geometric



thickness, the dust-induced cloud-top longwave warming response per unit DOD to increases in dust-base heights ($\frac{\partial LwHt_{cldtop}}{\partial DB}$) is also generally positive, although with varying magnitude for categories of dust-layer geometric thicknesses (Fig. 9c). On average, mean dust-induced cloud-top longwave warming responses per unit DOD to increases in dust-base heights (10.3% and 10.6%) are approximately the same for low and high DOD categories (Fig. 9d).

### 3.4.3 Contributions of dust-layer characteristics and optical depth to dust-induced radiative warming near cloud top.

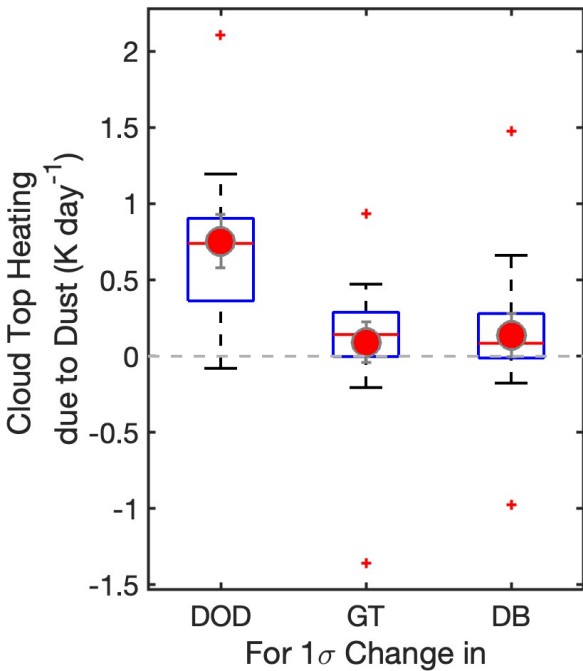

**Figure 10: Box-and-whisker plot showing dust-induced cloud top heating (K day⁻¹) for a 1σ (1 standard deviation) change in dust optical depth (DOD; 1σ = 0.27), geometric thickness (GT; 1σ = 0.964 km), and dust base height (DB; 1σ = 0.778 km). Red circles represent the mean values, with error bars indicating the standard error.**

The results above have shown that there are general increases in dust-induced longwave warming for a unit increase in dust-layer base height and geometric thickness, as well as dust optical depth. To understand the relative impacts of these parameters on the dust-induced cloud-top longwave warming response, we estimate the dust-induced cloud-top heating one per standard deviation change in each variable ($\sigma_X \frac{\partial LwHt_{cldtop}}{\partial X}$; where X is DOD, GT or DB and $\sigma_x$ is their standard deviations; compare Fig. 10 and Fig. 6). While one standard deviation increases in dust-layer geometric thickness or base height result



in increases in dust-induced longwave warming at cloud top, their contributions are smaller than those due to dust optical depth. The contributions of dust optical depth on cloud top dust-induced longwave warming are about 8 times compared to the contribution due to dust geometric thickness, and about 6 times compared to the contribution due to dust base height. Specifically, dust-induced cloud-top heating 860 increases by 0.75 (±0.17) K day$^{-1}$ for a one-standard-deviation increase in dust optical depth, while it increases by 0.09 (±0.13) K day$^{-1}$ and 0.13 (±0.14) K day$^{-1}$ per one-standard-deviation increase in dust geometric thickness and dust-base height, respectively (see Fig. 10).

### 3.4.4 Sensitivity to cloud-top longwave warming to dust absorption properties

While the contribution of dust optical depth to the dust-induced cloud-top longwave warming response is 865 stronger than other dust-layer characteristics (e.g., Fig. 10), the impact of dust optical depth sensitively depends on dust physicochemical and absorption properties, including the dust size distribution and dust complex refractive indices. Because the same dust optical depth can be achieved for different combinations of dust size distribution and refractive index, accurate representation of these properties becomes important to estimate dust-induced cloud-top longwave warming and its influence on low-level 870 cloudiness. In addition, because these dust properties are difficult to measure from remote sensing and our current understanding mostly relies on limited in-situ measurements, dust models used in radiative transfer retrievals (for example in CloudSat-CALIPSO radiative flux dataset) with limited representation of dust properties, such as using spatiotemporally invariant dust properties, are subject to large uncertainties that may bias our understanding of dust-cloud relationship. Therefore, to understand how 875 uncertainties in dust size distribution and refractive indices may influence retrieved heating rates and dust-induced cloud-top warming, we use the SBDART model (see Section 2.2.4) with a range of observed dust properties over the North Atlantic. Essentially, the radiative transfer simulations seek to recreate the CloudSat-CALIPSO heating rates as closely as possible by using the retrieved dust properties (such as DOD and extinction profiles) and cloud properties, as well as the thermodynamic profiles (Fig. S14) 880 provided in the same datasets, but vary the dust size distribution and complex refractive index (Fig. S4). Specifically, we consider three measurement-based particle size distributions representing dust at source regions (volume mean diameter, $D_{vm}$ 18.02 µm), mid-range transport (16.86 µm), and after long-range transport (3.36 µm) (Fig. S13a). The coarse-mode proportion decreases from around 89% near the source to approximately 50% for long-range transport. In addition, we use three different measurements of 885 refractive indices, including from Di Biagio et al. (2019), Optical Properties of Aerosols and Clouds (OPAC, Hess et al., 1998), and Volz (1972), with shortwave imaginary values of 0.0026, 0.0075, and 0.0080 at 0.5 µm (see Fig S4c) and longwave imaginary refractive indices at 10 µm, 0.3534, 0.4783, and 0.1620 respectively. These combinations yield nine cases with spectrally varying extinction efficiency, single-scattering albedo, and asymmetry parameters, which serve as inputs to SBDART to examine the 890 sensitivity of dust-induced heating to changes in size and absorption properties, to understand how the



unaccounted uncertainties in the dust absorption spectrum may affect dust-induced cloud-top warming (see Fig S4 e-f).

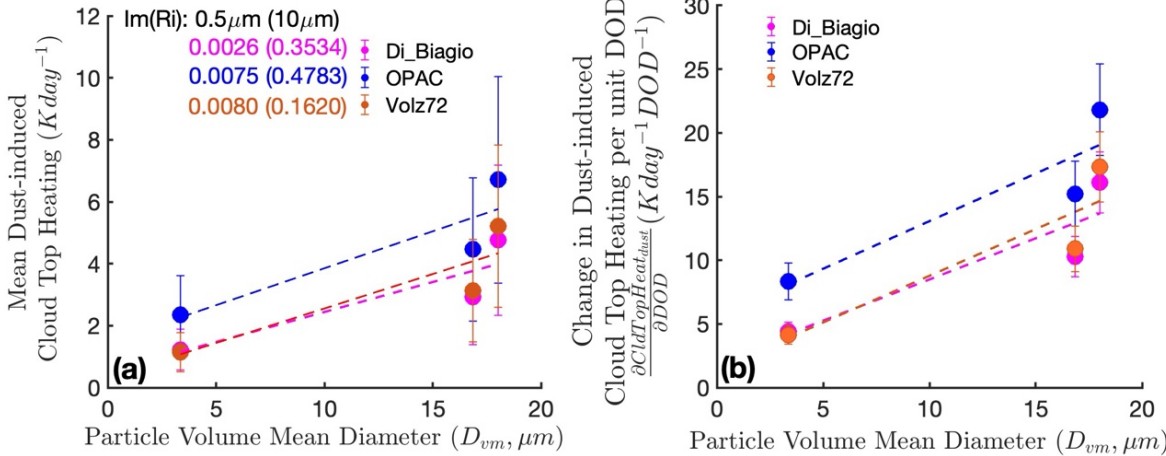

**Figure 11: (a) Mean cloud top heating due to dust ($CldTopHeat_{dust}a$) and (b) its sensitivity to dust optical depth $\left(\frac{\partial CldTopHeat_{dust}}{\partial DOD}; K\ day^{-1} DOD^{-1}\right)$ for different dust particle volume diameter and refractive index. The x-axis shows the particle volume mean diameter ($D_{vm}$, μm). Colors indicate refractive index sources: Di_Biagio (magenta), OPAC (Blue), and Volz72 (orange). Corresponding imaginary parts of refractive indices are shown Im(Ri) for wavelengths 0.5μm (and 10μm). Error bars denote standard deviation.**

The result shows that the cloud-top heating response to dust optical depth depends sensitively on dust size distribution and refractive indices (see Fig. 11). Specifically, near the dust source where the fraction of coarser dust particles is higher than after mid or long-range transport, the absorption of longwave radiation by these coarser particles results in stronger longwave cooling in the dust layer and thereby, a stronger dust-induced longwave warming at the low-level cloud top (Fig. 11a & Fig. S15). Between when the dust is emitted at the source and when they are transported, the fraction of coarse dust particles changes significantly, resulting in the dust-induced cloud-top heating significantly reducing along the dust transport pathway. Specifically, we find that cloud-top heating is roughly twice as strong for mid-range dust size distributions and nearly four times stronger for near-source dust size distributions, compared to those dominated by finer particles after long-range transport. For example, using refractive indices from Di Biagio, the corresponding dust-induced cloud-top heating rates are $4.76 \pm 2.42$, $2.93 \pm 1.54$, and $1.22 \pm 0.66$ K day$^{-1}$, for source, mid- and long-range transport, respectively (Fig. 12a), while the associated sensitivities per unit change in dust optical depth are $16.10 \pm 2.39$, $10.29 \pm 1.58$, and $4.4 \pm 0.75$ K day$^{-1}$ DOD$^{-1}$, respectively (Fig. 11b).



Similarly, dust layers with higher values of the imaginary part of long-wave refractive indices (at 10 μm wavelength) have stronger longwave cooling in the dust layer and stronger dust-induced longwave warming at cloud top, along with greater sensitivity per unit change in dust optical depth (see blue circles in Fig. 11a & 11b and blue bars in Fig. S15e). It is evident that while the dust heating rates depend on the spectral variation of the refractive index, the dust-induced changes in the low-level cloud top are primarily

controlled by the longwave part of the spectrum. In other words, stronger longwave absorption (represented at 10 μm) than shortwave absorption (represented at 0.5 μm) dominates the dust-induced changes at cloud top. For example, simulations using the long-range dust size distribution show the strongest cloud-top warming with OPAC ($2.35 \pm 1.26$ K day$^{-1}$), even though Volz72 has the highest value of shortwave imaginary refractive index at 0.5 μm wavelength (0.0080) (followed by OPAC, 0.0075, and

Di Biagio, 0.0026). In contrast, OPAC has higher imaginary refractive index of 0.4783 at 10 μm wavelength compared to Di Biagio and Volz72, with 0.3534 and 0.1620, respectively, thus producing weaker cloud-top warming ($1.22 \pm 0.66$ K day$^{-1}$ and $1.15 \pm 0.63$ K day$^{-1}$) (see Fig. 11a). Overall, our results indicate that dust-induced longwave warming at the cloud-top is sensitive to dust absorption properties, particularly the longwave imaginary refractive index and the fraction of coarse-mode particles,

which may not be accurately captured in the CloudSat-CALIPSO merged dataset that used spatially invariant dust absorption properties in the retrieval radiative fluxes.

### 3.4.5 Relationship between dust-induced longwave warming and low-level cloud response

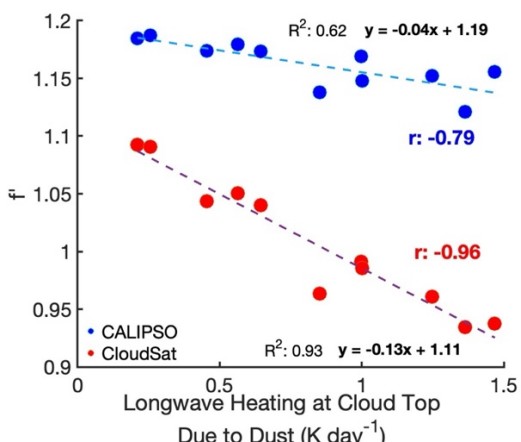

**Figure 12: Scatter plot showing changes in grid-normalized cloud fraction relative to dust-induced cloud top heating. Blue circles are when cloud fraction data is used from CALIPSO, while red dots are when cloud cover is used from CloudSat.**

Despite the sensitivity to dust absorption properties, our results have shown that the dust-induced cloud-top longwave warming responses to increases in dust optical depth, dust-base height, and geometric





thickness are generally consistent with similar responses of cloud fraction anomalies, although with different signs. Consequently, we find that decreases in grid-normalized cloud fraction anomalies correlate well with increases in low-level dust-induced radiative warming near the cloud top (Fig. 12). Specifically, this correlation between reduced cloud fraction and increased dust-induced longwave warming is about -0.96 for CloudSat and about -0.79 for CALIPSO. This suggests that dust-included
longwave warming at cloud top, which reduced the overall all-sky cloud-top longwave cooling, is primarily responsible for the decrease in low-level cloudiness when dust is above the cloud.

**3.5 Influence of Meteorology on the Cloud response to dust-layer Characteristics**

While we used a methodology that minimizes the influence of meteorology in our estimates, its influence
may still confound the relationship between dust-induced cloud-top longwave warming and the low-level cloud responses to dust-layer characteristics and dust optical depth. Such influence may come from the instantaneous assessment of the radiative heating rates since that depends on the background thermodynamic profile and/or the large-scale cloud-controlling environmental parameters, such as sea surface temperature and surface wind speed. We assess these influences on our results in the subsections
below.

**3.5.1 Sensitivity of dust-induced cloud-top radiative warming to background thermodynamics.**

We use SBDART to separate the influence of the background thermodynamic profile, including the influence of temperature and humidity, from the dust properties, and examine their contribution to the simulated radiative heating rates. Like above, we do so by recreating the CloudSat-CALIPSO- heating
rates as closely as possible, using the derived dust properties (such as DOD and extinction profiles) and cloud properties (such as cloud optical depth and cloud top heights), as well as the thermodynamical profiles (temperature and humidity) provided in the same datasets (see section 2.2.4 for details). The variations in the simulated dust-induced shortwave and longwave heating rates for each dust-layer category are similar, although with different magnitudes (because we use different size distributions and
refractive indices), as those obtained from the CloudSat-CALIPSO dataset (compare Fig. S9 and S13). Overall, the SBDART-simulated mean dust-induced cloud-top heating of 1.38 K day$^{-1}$ (calculated using dust optical depth, extinction profiles, cloud properties, thermodynamic profiles, and a long-range transported dust size distribution with Di Biagio refractive indices), is approximately 1.7 times higher than the value estimated from CloudSat-CALIPSO data which is 0.81 K day$^{-1}$. To enable comparison,
both SBDART-simulated and CloudSat-CALIPSO-derived heating values across categories were normalized by their respective means (see Fig. 13a; compare black circles with grey circles).





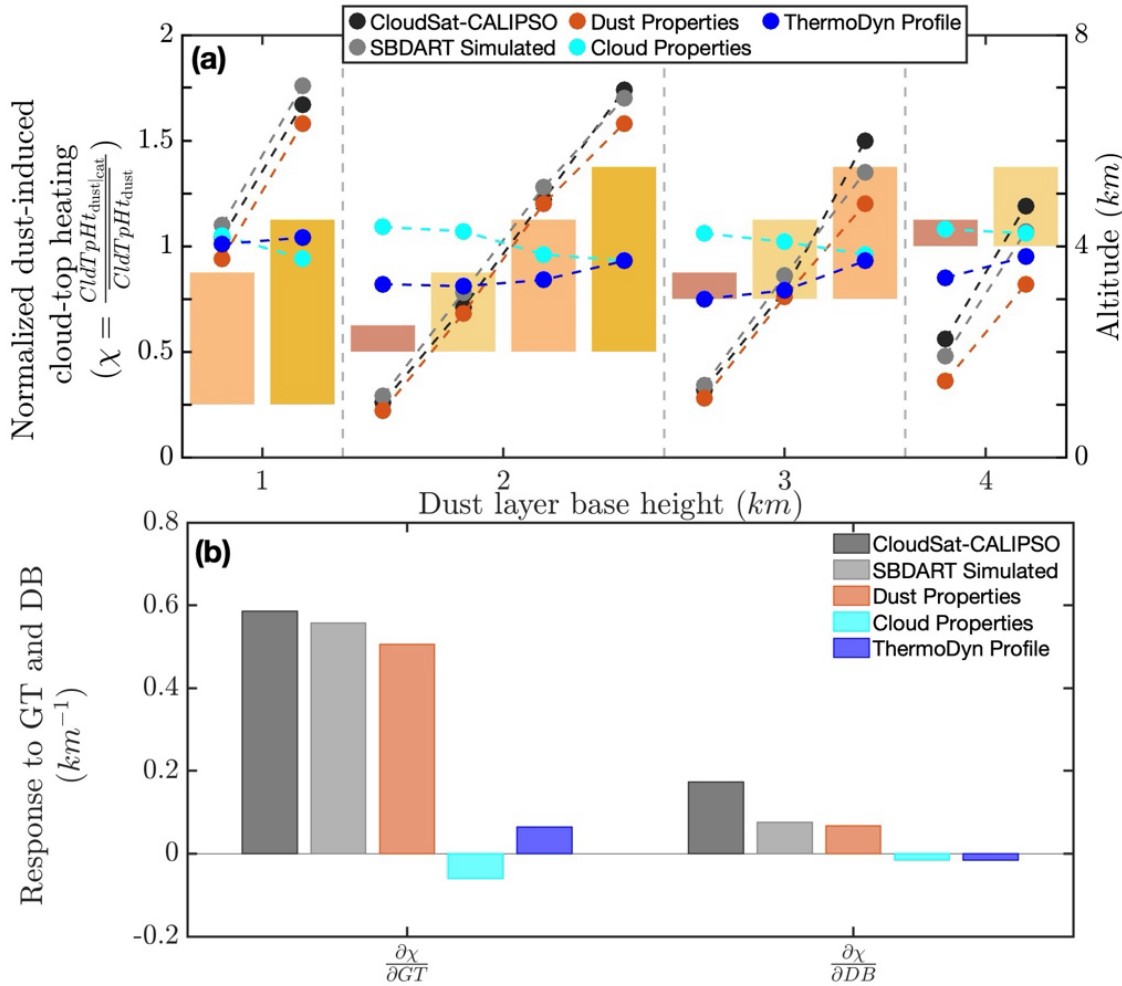

**Figure 13:(a) Normalized dust-induced cloud-top heating across dust-layer categories $\left(\chi = \dfrac{CldTpHt_{dust|cat}}{CldTpHt_{dust}}\right)$. Cloud-top heating derived from the CloudSat-CALIPSO dataset (black markers) is normalized by mean across categories (0.81 K day$^{-1}$), while SBDART-simulated heating (grey, orange, cyan, blue markers) is normalized by corresponding simulated mean across categories (1.38 K day$^{-1}$). Grey markers show simulations with all factors varying. Orange, cyan, and blue markers isolate the effects of dust properties, cloud properties, and thermodynamic profiles, respectively. Bars show the dust categories, with the right y-axis as altitude (b) Cloud-top heating response to a 1 km increase in dust layer thickness and base altitude. Black bars represent sensitivities derived from CloudSat-CALIPSO. Grey bars show the total SBDART-simulated response with all factors varying. Orange, cyan, and blue bars isolate the effects of dust properties, cloud properties, and thermodynamic profiles, respectively.**





Consequently, we estimated the sensitivity of the dust-induced cloud-top warming to variabilities in dust properties, cloud properties, and thermodynamic profiles by differencing with the simulation that replaces each set of parameters with their mean state (see Fig. S14 and Table S1). Therefore, with these
simulations, we can estimate the contributions of the variabilities in each parameter to the simulated dust-induced cloud-top heating (see section 2.2.4 for details). The result shows that the response of the cloud-top heating rate to dust-layer characteristics is less sensitive to the variabilities in the cloud properties and background thermodynamic profiles, thereby contributing less to the overall dust-induced cloud-top heating rate when compared to the variabilities in dust properties (Fig. 13b). The influence is despite the
co-variabilities in the cloud properties and thermodynamic profiles with the dust-layer properties, including dust-layer geometric thickness and dust-base height (e.g., Fig. S14). The responses of the normalized dust-induced cloud-top heating to a unit increase in dust-layer geometric thickness are $0.51\,\mathrm{km^{-1}}$ when only dust properties are varied (orange bars in Fig. 13b), $0.06\,\mathrm{km^{-1}}$ when only thermodynamic profiles are varied (blue bars in Fig. 13b), and $-0.06\,\mathrm{km^{-1}}$ when only cloud properties are
varied (cyan bars in Fig. 13b). Similarly, the responses of the normalized dust-induced cloud-top heating to a unit increase in dust-layer base height are $0.07\,\mathrm{km^{-1}}$ when only dust properties are varied, $-0.02\,\mathrm{km^{-1}}$ when only thermodynamic profiles or cloud properties are varied. This highlights the dominant role of dust properties in the estimated dust-induced cloud-top heating, and consequently, its relationship to the reduced cloud fraction when dust is above the cloud. Overall, our results show that dust-induced cloud-
top radiative warming is more sensitive to changes in dust properties, contributing more than 90% of the estimated dust-induced cloud-top warming, than to changes in cloud properties or thermodynamic profiles.

### 3.5.2 Influence of Other Meteorological Variables on cloud response to dust-layer characteristics

In addition to examining the sensitivity of our results to the variabilities in cloud properties and
thermodynamic profiles represented by the background temperature and moisture profiles, we also examined the influence of other cloud-controlling parameters that may be linked to large-scale atmospheric conditions. We focus on cloud-controlling parameters that have been identified by previous studies over the North Atlantic region, including sea surface temperature (SST), wind speed, lower tropospheric stability, and above-cloud humidity (Klein et al., 2017; Naud et al., 2023). To minimize the
influence of these cloud-controlling factors, we confined our analysis to $5°×5°$ grid boxes, where we assume the meteorological variability is small when focused on a single season for a long-term period. While our approach is designed to minimize the overall influence of meteorology and reduce the impact of possible co-variability between meteorological conditions and aerosols on the low-level cloud cover, we further examine how the other well-known cloud-controlling factors may change for each category
used in our analysis (e.g., Fig. 4).



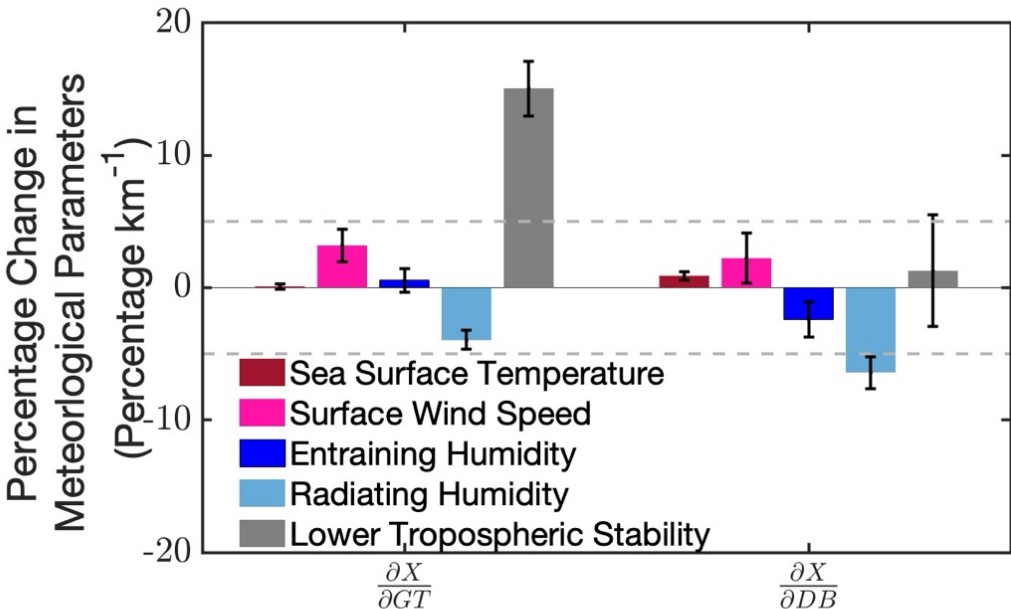

**Figure 14: Percentage change in key meteorological variables per kilometer increase in dust layer geometrical thickness and base height. Shown variables include sea surface temperature (SST, red), surface wind speed (magenta), entraining humidity (dark blue), radiating humidity (light blue), and lower tropospheric stability (LTS, grey). Percentage changes are computed relative to the domain-mean values of each variable: SST = 25°C, wind speed = 6.04 m s⁻¹, entraining humidity = 0.0077 kg/kg, radiating humidity = 0.0066 kg/kg, and LTS = 7.08°C. Error bars are the standard error for respective parameters. Grey dotted horizontal lines are 5% lines.**

We find that the variations are minimal for most of these cloud-controlling variables across our defined categories, suggesting that these factors have little impact on the cloud response to dust-layer characteristics (see Fig 14 & Fig. S16). Like percentage cloud response (Fig. 4b &c) and cloud top heating response (Fig. 9b&c) to per one-kilometer increase in GT and DB, we estimated percentage changes for relevant meteorological parameters to GT and DB changes (see Fig 14). Most of the meteorological variables that can affect the cloud cover show less than 5% changes to increase in GT and DB. For example, a kilometer increase in GT is associated with only a 0.54% increase in SST (red bars in Fig 14) and a 3.1% increase in surface wind speed (magenta bars in Fig 14). In the case of DB, these changes are, 0.9% for SST and 2.2 % for windspeed (see red and magenta bars in Fig 14). Similarly, the above-cloud moisture has minimal impacts on the low-level cloud response to dust-layer characteristics. As shown in Fig. 13, meteorological impacts on the response of cloud-top dust-induced longwave warming to dust-layer characteristics were small despite the mid-tropospheric high relative humidity used in the radiative transfer simulations (see also Fig. S16). To show further the minimal impacts of above-cloud moisture in our analysis, we estimated two grid-scale humidity measures following Eastman and Wood (2018):





entraining humidity (adjacent to the cloud top) and radiating humidity (the average humidity between 700 hPa and the entraining layer, as discussed in Section 2.2.6). Radiating humidity emits downwelling

longwave radiation, affecting cloud-top cooling (Yamaguchi et al., 2015), while higher entraining humidity reduces droplet evaporation, lowering the likelihood of cloud breakup. We find both entraining and radiating humidity anomalies change very little across categories (see Fig. S16c & S16d), similar to changes in SST and wind speed anomalies. Specifically, the changes in entraining and radiating humidities per one-kilometer increase of GT are about 3% and for DB it is below 6 % of the domain mean

values, respectively (see blue bars in Fig. 14). Overall, the changes in sea surface temperature, surface winds, and above-cloud moisture cannot explain the large changes in low-level cloud response to unit increase in dust-layer characteristics and properties.

In contrast to wind speed, SST, and above-cloud moisture anomalies, the change in LTS is substantial

across the dust categories (Fig. S16e). Specifically, we find that for per kilometer increase in GT, LTS anomalies increase by 15% (first grey bar in Fig. 14), while for DB increase LTS change is 1.27% (second grey bar in Fig. 14). This result is consistent with previous studies of dust above clouds, where increases in LTS attributed to shortwave warming in the dust layer have been associated with increases in low-level cloudiness (Doherty and Evan, 2014). This relationship between LST and enhancement of low-level cloud

cover was often explained by boundary-layer moisture build-up, resulting in increases in cloud-top longwave cooling and consequently increases in cloud fraction (e.g., Wilcox et al., 2009). However, our results show an observed reduction in low-level cloud fraction associated with increasing dust-induced cloud-top warming when the dust-layer geometric thickness and optical properties are increased. Our results further suggest that dust-induced longwave warming at the cloud top may offset the influence of

LTS on the mean cloud-top cooling, and, while LST may be important for low-level cloud enhancement and dust semi-direct effect, the influence of LST may only be stronger in low dust condition, with less coarse dust particles and less overall longwave absorption.

## 4. Discussion and Summary

We have shown that low-level cloud response depends sensitively on the dust-layer altitude, geometric

thickness, and dust optical depth, influencing the dust-induced longwave warming and counteracting the mean radiative cooling near the low-level cloud top. We do so by leveraging vertically-resolved aerosol, cloud, and radiative heating rates information from CALIOP and the merged CALIPSO-CloudSat products, as well as the Santa Barbara DISORT Atmospheric Radiative Transfer (SBDART) model between May and August (2007 – 2017) over the tropical North Atlantic Ocean (55W-20W; 5N-30N).

During this period, when the mean low-level cloud top height is approximately 1.35 km (± 0.05 km), we showed that nearly 78% of the total dust column burden is situated above the cloud layer. We use profiles with CloudSat-CALIPSO retrieved dust layer and single-layer low-level clouds and require a minimum



of 200 m between them to minimize potential biases that can be introduced through uncertainties in the retrieved layer information. Furthermore, we also minimize large-scale meteorological variability by dividing the broad region into a 5°×5° degree grid box following previous studies (Adebiyi and Zuidema, 2018; Loeb and Schuster, 2008). We categorized dust and cloud profiles within each 5°×5° grid box based on the dust layer base height (1, 2, 3, and 4 km) and geometrical thickness (0.5, 1.5, 2.5, and 3.5 km) only when aerosols, clouds, and radiative flux information are all available to obtain statistically robust relationships. Out of the possible 16 categories of dust-layer base height and geometric thickness, only 11 satisfy our stringent requirements of the minimum number of profiles within each category and the minimum number of total 5°×5° grid boxes allowed.

Whereas low-level cloudiness typically increases due to the presence of the above-cloud dust layer through the process of dust semi-direct effect (Amiri-Farahani et al., 2017), we find that the extent of such increases depends on the dust layer's optical properties as well as its relative altitude and geometrical thickness. Using the above dust-layer categories, our results indicate decreases in low-level cloud response, that is, the rate of change of grid-normalized low-level cloud fraction relative to basin-wide mean per increase in dust-base height, geometric thickness, and dust optical depth. Specifically, our results suggest the following: First, the low-level cloud response is strongly influenced by both the altitude of the dust layer base and its geometric thickness. On average, cloud fraction decreases by 1.7% and 5.1% for each kilometer increase in dust-layer geometric thickness based on CALIPSO and CloudSat, respectively. While the cloud response to geometric thickness consistently leads to a reduction regardless of dust base height, the grid-normalized low-level cloud response to a one-unit change in dust-base height depends on the dust-layer geometric thickness. Specifically, for thinner dust layers, one unit increase in dust-layer base height reduces cloud cover by 0.5% and 1.4% for geometric thickness values of 0.5 km and 1.5 km, respectively, using CALIPSO data and 2.3% and 3.0%, respectively, using CloudSat. In contrast, for thick dust layers (GT=3.5 km), increasing the base height leads to an increase in cloud response, with values of 3.2% and 0.3% from CALIPSO and CloudSat data, respectively.

Second, we estimate cloud response to dust optical depth and found that the mean low-level cloud fraction decreases by approximately 17% for every unit change in dust optical depth. Consequently, when we divided the dust-layer categories into low and high dust optical depth cases, the low-level cloud response to dust-layer geometric thickness is stronger for high dust optical depth, despite having lower grid-normalized cloud fraction, than for low dust optical depth cases Specifically, for a 1 km increase in dust-layer base, the negative cloud responses change by about 0.7% (-0.1% and 0.6%, respectively) between the low and high dust optical depth categories. Similarly, for a 1 km increase in dust-layer geometric thickness, the negative cloud responses change by about 1.6% (-0.86 % and -2.49%, respectively) between the low and high dust optical depth categories. Finally, we estimate the relative contribution of the dust-layer characteristics and dust optical depth by estimating the sensitivity in the low-level cloud response



to a one-standard-deviation (1σ) increase in each parameter. We find that the negative low-level cloud response is stronger for a 1σ change in dust optical depth than for a 1σ change in dust-layer geometric thickness or dust-base heights. Specifically, we find that the relative reduction in low-level cloud response to 1σ increase in dust optical depth is about -4.3 ($\pm$1.04) %, compared to -1.6 ($\pm$ 0.65) % and 0.19 ($\pm$0.45) % to 1σ change in dust-layer geometric thickness and dust base respectively, indicating that the changes in low-level cloud fraction due to 1σ change in dust optical depth is about three times stronger than 1σ change in geometric thickness, and significantly more than 1σ change in dust-base height that is largely negligible within the error range.

We hypothesized that the variations in radiative fluxes at the low-level cloud top drive the observed cloud responses to perturbations in dust-base heights, geometric thickness, and dust optical depth. To show this relationship, we used the CALIPSO-CloudSat combined product to estimate radiative heating rates (both shortwave and longwave) for each defined dust category. We find that the dust layers above the low-level clouds induce anomalous longwave warming near the cloud top that counteracts the mean cloud-top radiative cooling and can reduce the net cloud-top cooling by as much as 18.7% (mean of 9.3%). In addition, our results further showed that this dust-induced cloud-top longwave warming increases with an increase in dust optical depth, dust-layer geometric thickness, and dust-base height. However, a 1σ increase in dust optical depth results in a significantly stronger dust-induced cloud-top warming ($\sim$0.75 $\pm$ 0.17 K day$^{-1}$), nearly 8 times greater than the warming from a 1σ increase in geometric thickness (0.09 $\pm$ 0.13 K day$^{-1}$) and 6 times greater than the warming from a 1σ increase in dust base height (0.13 $\pm$ 0.14 K day$^{-1}$). Consequently, we find that decreases in low-level cloud fraction response correlate well (0.79 for CALIPSO and 0.96 for CloudSat) with responses in dust-induced radiative warming near the cloud top. This suggests that dust-induced longwave warming, which reduced the overall cloud-top longwave cooling, is primarily responsible for the decrease in low-level cloudiness when above-cloud dust-layer characteristics and dust optical depth are increased.

While the contribution of dust optical depth to the dust-induced cloud-top longwave warming response is stronger than other dust-layer characteristics, our results further show that the impacts of dust optical depth on low-level clouds sensitively depend on dust absorption properties, including the dust size distribution and dust complex refractive indices. To understand how uncertainties in absorption properties may influence the estimated dust-induced cloud-top warming and consequently the low-level cloud fraction, we use the SBDART (Santa Barbara DISORT Atmospheric Radiative Transfer) model to recreate the CloudSat-CALIPSO heating rates as closely as possible by using the retrieved dust properties (such as dust optical depth and extinction profiles) and cloud properties, as well as the thermodynamic profiles provided in the same datasets, but vary the dust size distribution and complex refractive index based on previously-published measurements. The result shows that dust-induced longwave warming at the cloud top is highly sensitive to the coarse-mode fraction of the dust size distribution and the imaginary



part of the longwave refractive index. For example, near-source and mid-range-transported dust size
distributions containing approximately 89% and 83% coarse-mode fraction produce nearly four and two
times more warming, respectively, than one long-range-transported dust size distribution with only 50%
coarse-mode fraction. Similarly, higher values of the longwave imaginary refractive index, which indicate
stronger dust-layer longwave absorption, also lead to enhanced dust-induced cloud-top warming.
Therefore, our result suggests that the contribution of the dust-induced cloud-top longwave warming that
counteracts the mean cloud-top radiative cooling may be larger than indicated above, if the
spatiotemporally varying dust size distribution and refractive index are accurately represented.

While we minimize the influence of potential meteorological co-variability in our analysis, we further
assess its impact on the relationship between dust-induced cloud-top longwave warming and the low-
level cloud response. First, similar to the sensitivity analysis above, we also reproduced CALIPSO-
Cloudsat heating profiles using SBDART and performed sensitivity experiments by varying dust-layer
properties, cloud properties (including the cloud optical depth and cloud top heights), and thermodynamic
profiles (represented by the temperature and humidity profiles). The result showed that the estimated low-
level cloud response when dust is above the cloud can be primarily attributed to dust-layer characteristics
and properties, which dominate changes in dust-induced reduction in cloud-top longwave cooling.
Although the response of the cloud-top heating rates to dust-layer characteristics also co-vary with cloud
properties and the background thermodynamic profiles, specifically the changes in humidity profile above
the clouds, their contributions to the overall dust-induced cloud-top heating rate are less compared to the
dust properties. Second, we also examined the influence of other cloud-controlling parameters, including
sea surface temperature, wind speed, and above-cloud moisture that may be linked to large-scale
atmospheric conditions. The result showed minimal and non-systematic variations in these parameters
across dust categories, suggesting that they have limited or no influence on the observed relationship
between low-level cloudiness and anomalous dust-induced cloud-top longwave warming. Although we
find an increase in LTS for higher geometrically and optically thick layers, consistent with previous
studies, they contradict the observed decrease in low-level cloudiness for the dust categories. Overall, our
results indicate that the relationship between low-level cloudiness and dust-induced cloud-top radiative
warming is more sensitive to changes in dust properties than to meteorological variabilities.

Our results have important implications for our understanding of low-level cloud response and the dust
semi-direct effect over the North Atlantic Ocean. First, our result suggests that processes associated with
low-level cloud response when the dust layer is above the cloud and, consequently, the dust semi-direct
effect, are more sensitive to changes in cloud-top radiative cooling. Previous studies have shown the
significance of cloud-top radiative cooling in driving the cloud-top turbulence, convective instability,
mixing in the cloud layer, and small-scale dry air entrainment, which is important for cloud development
(Wood, 2012; Zuidema et al., 2019). A reduction in cloud-top radiative cooling could facilitate a reduction





in cloud fraction through several different pathways. One such pathway is that a reduction in cloud-top radiative cooling may lead to weaker turbulent mixing within the cloud, potentially resulting in decoupling and stratification of the boundary layer, which limits the surface moisture source into the cloud layer and subsequently reduces cloudiness (e.g., Bretherton and Wyant, 1997). Such pathways tend to result in an increase in the frequency of cumulus clouds rather than stratus clouds. Another pathway is through the entrainment of dry air at the cloud top, whereby a reduction in cloud-top radiative cooling will result in the weakening of the entrainment interfacial layer, allowing more dry and warm air from the free troposphere to mix into the cloud layer, thereby leading to reduced cloudiness (Wood, 2012b). For example, Wood (2012) highlighted that because a lower surface-based lifting condensation layer than the cloud-top inversion layer is required to maintain low-level cloudiness, 1 K of warming within the cloud layer aided by reduction of cloud-top radiative cooling or associated entrainment of dry air could lift condensation layer much faster than it lifts the cloud-top inversion layer, hence resulting in a reduction in cloudiness. Assuming other critical environmental factors remain constant, these pathways could result in a feedback process where a reduction in cloudiness can result in a further reduction in cloud-top radiative cooling and a further reduction in cloudiness (Bretherton, 2015; Ceppi et al., 2017). Such feedback relationships have been used to explain the stratocumulus to cumulus transition, which also exists over the North Atlantic Ocean (Sandu and Stevens, 2011). Our result, therefore, suggests that dust-induced longwave warming at the cloud top facilitates the reduction of low-level cloudiness, and those changes are strongly dependent on the dust layer's height, geometric thickness, and optical depth as well as dust absorption properties, including dust size distribution and refractive index.

Second, our result of radiative changes and cloud responses to the above-cloud dust layer adds to the traditional process associated with cloud responses to other above-cloud absorbing aerosols, such as biomass-burning smoke aerosols above low-level clouds of the southeast Atlantic Ocean. Our results highlight that in addition to the shortwave absorption that has been traditionally associated with low-level cloudiness and its aerosol semi-direct effect, the longwave effect could also play a substantial role. Specifically, the shortwave warming by the dust layer likely increases the lower-tropospheric stability for optically and geometrically thicker dust layers, similar to smoke aerosols, where such increased stability would enhance low-level cloudiness (e.g., Wilcox, 2010, 2012). However, unlike smoke aerosols, the dust-induced longwave warming that reduces the mean cloud-top radiative cooling can affect cloud development and reduce low-level cloudiness, similar to the radiative effects of above-cloud moisture, carbon dioxide, and free-tropospheric clouds on low-level clouds, as shown by previous studies(Adebiyi et al., 2015; Adebiyi and Zuidema, 2018; Christensen et al., 2013; Singer and Schneider, 2023). This interplay between shortwave and longwave effects on the clouds defines the dust semi-direct effect, where cloud responses play a pivotal role in deciding whether the overall effect will be dominated by the radiative warming (positive SDE) or cooling (negative SDE) at the top of the atmosphere. Specifically, the stability effect induced by the shortwave warming within the dust layer tends to increase low-level



cloudiness, resulting in negative dust SDE, but counteracted by the dust-induced longwave warming that reduces the cloud-top cooling and may suppress the stability effects of shortwave absorption, therefore, leading to reduction the low-level cloudiness and tipping the balance toward positive dust SDE. The net dust SDE, therefore, emerges as the outcome of competing longwave and shortwave effects. Our analysis shows that the relative contributions of these effects depend on the vertical positioning of the dust layer relative to the cloud and the layer's optical and geometrical properties.

We note some caveats in our analysis. First, our study relies on instantaneous snapshots from CALIPSO-CloudSat profiles, which do not capture the full temporal dynamics, including diurnal variation, of the low-level clouds over the region. In addition, while these profiles provide vertical distributions of clouds and aerosols—suitable for our analysis and useful to compare with snapshots from passive sensors like MODIS—the CALIOP sensors have limited capability in detecting aerosols below clouds(Yu et al., 2015). As a result, the selected profiles for the above-cloud dust cases might not be entirely free from the influence of aerosols below the cloud. This limitation introduces uncertainty, as aerosols below the cloud could affect cloud cover by altering cloud microphysics and, subsequently, the overall cloud response. Such aerosols may influence processes like droplet formation, cloud lifetime, and cloud cover, thereby complicating the interpretation of our results. However, this uncertainty cannot be addressed using the datasets and methodology employed in this study, and further research using numerical models may be required to accurately quantify these effects. Second, although we found minimal and non-systematic variation across the categories in meteorological cloud-controlling factors, highlighting their non-significant effect on our analysis, their role cannot be fully quantified. Specifically, the dust layer over the North Atlantic, similar to smoke over the South Atlantic, has been found to contain relatively higher moisture content (Ryder, 2021). Thus, the uncertainty related to above-cloud humidity remains a critical factor, although we attempt to account for this by using the radiating and entraining humidity parameters (Fig. 12).

Despite these limitations, our result underscores the need for accurate estimates of the dust layer's vertical distribution and dust absorption properties, including dust size distribution and mineralogical composition, to accurately quantify dust SDE. Unlike above-cloud smoke, where shortwave radiative effects often suffice to explain the SDE, dust requires consideration of both shortwave and longwave influences (Baró Pérez et al., 2021; Deaconu et al., 2019; Wilcox, 2010, 2012). The longwave effect, in particular, depends on dust vertical profiles, the thickness of the dust layer, and their altitude relative to the clouds, along with particle size distributions and absorption properties. Current climate models do not accurately incorporate these parameters, leading to uncertainties in the estimation of dust's SDE (Cappa et al., 2016), especially studies found the underestimation of large dust particles by current climate models, which may lead to potential underestimation of the longwave effect (Adebiyi et al., 2023; Adebiyi and Kok, 2020). Our study highlights the need for improved observation of global three-dimensional dust



distributions, along with accurate estimations of radiatively sensitive aerosol properties, such as particle size distribution and mineralogical composition.

**Data availability.** Cloud and aerosol layer data used in this study were obtained from the CALIPSO Level 2 data products, including the 5 km aerosol layer product
(https://doi.org/10.5067/CALIOP/CALIPSO/LID_L2_05KMAPRO-STANDARD-V4-20) and the 5 km aerosol profile product (https://doi.org/10.5067/CALIOP/CALIPSO/LID_L2_05KMALAY-STANDARD-V4-20). CloudSat-CALIPSO radiative flux data were accessed from the 2B-FLXHR-LIDAR product (https://www.cloudsat.cira.colostate.edu/data-products/2b-flxhr-lidar). All data supporting the findings of this study are available at Zenodo: https://doi.org/10.5281/zenodo.15571618.


**Supplement.** Attached is a supplement document containing additional figures and a table.

**Author Contributions.** AAA and SKP designed the study. SKP performed the analysis. SKP and AAA
interpreted the results and wrote the paper.

**Competing Interests.** The contact author declared that none of the authors has any competing interests.

**Acknowledgement.** This study was supported by the U.S. Department of Energy (DOE), Office of
Science (award #DE-SC0024281). CALIPSO data used in this study were obtained from the NASA Langley Research Center Atmospheric Science Data Center (https://asdc.larc.nasa.gov/project/CALIPSO), and we acknowledge the CALIPSO science team for providing these data. CloudSat-CALIPSO radiative flux data were provided by the CloudSat Data Processing Center at Colorado State University (https://www.cloudsat.cira.colostate.edu/), and we thank
the CloudSat team for their data processing and dissemination efforts. We acknowledge the use of the SBDART radiative transfer model, developed at the University of California, Santa Barbara, and made publicly available at https://github.com/paulricchiazzi/SBDART.

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
