# Peer review of "Dust semi-direct effects: The influence of free-tropospheric dust-induced longwave radiation on low-level cloud response over the North Atlantic Ocean"

_EGUsphere, 2025_

## Author Comment (AC1)

**Response to Reviewer #1**

The authors evaluate the semi-direct effect of dust aerosol over low-levels clouds in the North Atlantic Ocean with CloudSat radar and CALIOP lidar observations. They show that the summer free-tropospheric dust layer overlying low clouds induces a shortwave heating response which strengths the boundary layer inversion, consistent with previous studies of biomass burning aerosols (positive semi-direct effect). But also, due to the large dust particle sizes, a significant longwave warming response. The dust-induced longwave warming dominates the heat budget and leads to overall less cloud-top cooling (~10%) which reduces cloud cover (novel negative semi-direct effect).

The paper presents an interesting new finding about the longwave semi-direct effect of aerosols on low-level clouds. The authors discuss first the response of cloud cover, and then the response of heating rates, to dust optical depth, dust layer geometric thickness, and dust-layer base altitude. They use a radiative transfer model (SBDART) to quantify the cloud sensitivity to perturbations in each of these aspects, as well as meteorological quantities.

Overall, the results are interesting. However, I have some confusion about the methods used for the analysis, particularly the approach taken to remove confounding effects of meteorology and how to separate the impact of each metric of the dust layer.

We thank the reviewer for showing interest in our manuscript and for their thoughtful and constructive comments. We have responded to the points raised and incorporated the valuable suggestions, which have significantly improved our manuscript. We address (in blue font) each of the reviewer's comments (in black font), along with suggested changes in the manuscript, which are shown in italics.

**Major Comments**

**1.** Mainly, why have you chosen to split up the data into so many "categories" and compute the partial derivatives of interest as differences between categories instead of computing them with a multiple linear regression?

Thanks for the comment. Our primary goal was to isolate the sensitivity of cloud-top radiative cooling to different dust-layer properties (optical depth, geometric thickness, base height), cloud properties, and thermodynamic structure. We explored multilinear regression in the early stages of our analysis. However, because some of the dust-layer properties are correlated, accounting for the interaction terms (that is, cross-correlation terms between the correlating variables) significantly affects our interpretations. In addition, there are likely nonlinearities in the relationship between low-level cloud response and dust properties that are beyond the scope of this study, as well as the negative impacts of potential outliers, making it unsuitable to use a multilinear regression approach. Therefore, to avoid these problems and to better control the co-variability in the dust-layer properties, we binned the data into categories. Thus, our categorical

method allowed us to systematically vary one parameter set at a time, while holding the others fixed, enabling us to directly quantify their related cloud response. In addition, this also allows us to carry out corresponding radiative transfer calculations by varying combinations of aerosol, cloud, and thermodynamic structure and to quantify their sensitivities. To clarify, we have added the following text in the manuscript.

*"We used this categorical method to account for potential cross-correlation between predictors (such as dust geometric thickness, base height, and optical depth), which would otherwise require calculating interaction terms in a multilinear regression, and to avoid potential nonlinearities and negative impacts of outliers, allowing us to isolate the impact of each parameter while maintaining statistical robustness through sample size criteria."*

**2.** For the meteorology, why do you choose to analyze each 5×5° box separately rather than e.g. extending a cloud controlling factor analysis to include the dust properties of interest? Wouldn't this better remove any confounding meteorological factors, rather than just looking at some "small area"? Because surely there will be some variability still within 5×5° boxes.

As done in previous studies, we chose $5° \times 5°$ grid boxes to minimize the influence of cloud-controlling factors on our analysis. This is because our study focuses on instantaneous interactions between clouds and aerosols, and as such, this approach minimizes any potential memory that may be introduced into our statistical relationship due to the influence of large-scale meteorology (Mauger and Norris 2007). In addition, choosing a larger domain of analysis implies comparison of cloud/dust characteristics of entirely different locations. For example, the eastern parts of the domain, often dominated by dust that is relatively near to source regions, compared to those which are dominated by long-range transport, far from sources, are the western part of the domain. Furthermore, underlying meteorological conditions such as sea-surface temperature, wind speed, etc., significantly differ from east to west. Therefore, confining our analysis within 5°×5° boxes helps reduce such stark changes. Finally, for meteorology variables in Section 3.5.2, we also adopt a similar approach to ensure consistency with other parts of our analysis

**3.** For section 2.2.2 why do you split just into "low" and "high" dust optical depth? Why not have a continuous variable for the DOD?

Thank you for the suggestions. We use these broad categories, in part, because DOD distribution often skews towards low DOD (indicating that clean-to-low DOD occur far more frequently than high DOD cases), and to have relatively sizable and equitable sample sizes to compare, we use low and high DOD as representatives of broad aerosol conditions. In addition, and similar to our response above, this approach also aids the interpretability of our results and avoids the potential outlier effect. Yes, we acknowledge that binning into "low" and "high" optical depth reduces the number of available datasets significantly. However, it was necessary to compare the influence of a dust layer of the same geometric thickness, having differing optical depth, so we can quantify the influence of geometric thickness independent of optical depth. Ideally, we would have taken

more DOD bins; however, due to limited profiles, it was not possible to have enough samples available for each category and DOD bin to make a statistically significant average and comparison across categories. In addition, to quantify the influence of DOD. That said, the reviewer may be interested in Fig. S8, which used DOD from all available profiles for each geometric-thickness and dust-base combination to calculate the relationship between cloud fraction and DOD (see Fig. S8).

[Figure]

**Figure S8**: Change in $f'$ per unit change in DOD as a function of dust base height and geometric thickness

**4.** If instead of making these artificial categories and thresholds you used the continuous variables, you probably would be able to retain more data and have more statistical significance (instead of throwing out so many profiles that don't meet the minimum 180 per category). If you are concerned that some of these relationships may be quite nonlinear, so a linear regression wouldn't be suitable, maybe you could consider something like ln (DOD) instead of DOD directly.

We appreciate this suggestion. Part of our motivation for categorization was to avoid the inherent cross-correlation among predictors such as dust geometric thickness, dust layer height, and optical depth. In addition, while previous studies have indeed used ln (DOD), such characterization is currently not clear for dust geometric thickness and dust layer height. In addition, we used a minimum sample threshold for each category to ensure statistical significance and representativeness, although some profiles were excluded. That said, we also consider other minimum sample thresholds and our selection of at least 180 per category does not significantly impact the conclusion presented in the paper. In addition, this methodology also simplified the associated radiative transfer simulations using SBDART, as the corresponding values of thermodynamic profiles, dust-layer properties, and cloud properties could be directly used to perform the sensitivity analysis.

**Minor Comments**

**1:** L101/102: I think you have swapped reference to Fig 1c and 1d. The figure caption says that 1c is showing change in cloud cover when all aerosols are present and 1d is for when only dust is present.

Thank you for pointing this out. We have corrected the figure references accordingly.

**2:** L170: How can the dust "coexist with" but also be "typically above" the cloud layer?

Thank you for pointing this out. We have revised the sentence to read: "frequently overlies." Our original use of "coexist" referred to co-occurrence within the same atmospheric column, not mixing of dust and cloud in the same layers. This has now been clarified and modified sentence reads

*"We focused our analysis on the North Atlantic Ocean between May and August (2007-2017) when the SAL dust **frequently overlies** low-level cloud layers"*

**3:** L548: Is this a good estimate? What if the clouds are not stratiform and pushing through the inversion layer?

Thank you for pointing this out. We agree that identifying the inversion layer based solely on cloud-top height has limitations, particularly if clouds are not strictly stratiform. However, mentioned, we defined the inversion top as the layer immediately above the cloud top, as our interest here is not in diagnosing the detailed inversion structure but in capturing the humidity that could entrained into the cloud layer and change cloud parameters. For this purpose, the simplified definition of the inversion top as the layer just above the cloud top (following Eastman and Wood, 2018) consistent. We have clarified this

*"Though this is a simplified definition of the inversion top and may not fully apply to clouds with cumulus characteristics, here, we are mainly interested in moisture available for entrainment into cloud."*

**4:** L1059/1065/1066: "LST" should be "LTS" for lower-tropospheric stability.

Thank you for pointing this out. We have corrected this.

**Reference**

Mauger, G. S. and Norris, J. R.: Meteorological bias in satellite estimates of aerosol-cloud relationships, Geophysical Research Letters, 34, 1–5, https://doi.org/10.1029/2007GL029952, 2007.

Eastman, R. and Wood, R.: The competing effects of stability and humidity on subtropical stratocumulus entrainment and cloud evolution from a Lagrangian perspective, J Atmos Sci, 75, 2563–2578, https://doi.org/10.1175/JAS-D-18-0030.1, 2018.

---

## Author Comment (AC2)

Response to Reviewer #2

In my opinion, this is a long, but good paper that presents a thorough study examining several aspects of dust influences on underlying low-level clouds. My overall sense is that the topic is of interest to the community, the methodology is sound, and the presentation is of high quality. Nonetheless, I feel that the manuscript needs some refinement before publication. The necessary improvements seem fairly minor and include mainly some clarifications, elaborations on a few points, and wording changes. Please find my specific comments below.

We thank the reviewer for their time, encouraging remarks, and constructive comments. We are pleased that the reviewer finds the topic of interest to the community and appreciates the quality of the study. We have carefully considered all suggestions and made the corresponding changes to the manuscript, which have substantially improved the quality of the manuscript. Below, we provide a detailed response (in blue font) along with the reviewer's comments (in black font), and modifications in the manuscript are shown here in italics.

**Minor issues about content**

1. Line 222: It could help some readers if the manuscript briefly explained the relationship between depolarization and particle sphericity and briefly discussed the reason why the depolarization ratio differs for different aerosol populations (e.g., non-sphericity).

   We appreciate the suggestion and have modified it in the revised manuscript, which now reads:

   *Among these aerosol types, the identification of dust types (polluted dust, dusty marine, and pure dust) mainly depends on the estimated particle depolarization ratio, which quantifies the degree to which the light scattered by atmospheric particles undergoes depolarization.* ***Polarization refers to the orientation of the electric field vector of light waves, which can be altered upon scattering by particles. This ratio is linked to particle shape; spherical particles (such as marine aerosols or spherical droplets) have smaller effects on the polarization of incident light, resulting in low depolarization ratios, while non-spherical particles like mineral dust modulate the polarization more strongly, leading to higher ratios.*** *Specifically, layers with a high particulate depolarization ratio estimated to be greater than 0.20 are classified as pure dust, regardless of surface type, location, altitude, or integrated attenuated backscatter values****, owing to predominantly non-spherical particles.*** *Conversely, moderately depolarizing particles, with a depolarization ratio ranging from 0.075 to 0.20 and a layer base height below 2.5 km when observed over the ocean, fall into the category of dusty marine particles.*

2. Lines 330-334: It could help to explain somewhere why there is no sensitivity to humidity: Does this indicate that dust is not too hygroscopic?

   Thank you for pointing this out. We have added a sentence in Section 3.5.1 clarifying this aspect, which reads:

*"The lower sensitivity to the thermodynamic profile may partly result from the limited influence of moisture, despite higher relative humidity within the dust layer (see Fig. 2c and Fig. S14), as dust particles are largely non-hygroscopic, meaning their size and optical depth do not substantially increase with humidity."*

3. Line 378: It would help to clarify the text "meteorological variability is roughly invariant": What is meant by variability being invariant? The same applies to Lines 535-536.

   The sentence has been revised to clarify the intended meaning:

   *"One method for minimizing the meteorological influence on aerosol-cloud interactions is to limit the spatial domain, under the assumption that meteorological conditions, including large-scale circulation, do not vary significantly within the selected small region"*

4. Section 2.2.4: This section should specify somewhere the solar zenith angle (s) used in the SBDART simulations. Are the calculated radiative fluxes for the local times of CALIPSO-CloudSat overpasses or are they daily average values? This is quite important for the shortwave simulations mentioned in Line 480 and elsewhere in the manuscript.

   We thank the reviewer for highlighting this important clarification. We have updated Section 2.2.4 to include the following:

   *"These simulations were conducted using solar zenith angles corresponding to the approximate local time of CALIPSO overpasses (~15:00 UTC) in the study region. As such, the radiative fluxes represent near-instantaneous conditions at the time of satellite observations, not daily averages."*

5. Section 2.2 or 3.1: I wonder if the impact of dust layer base altitude partly comes from the fact that if dust base altitude is low, only clouds at very low altitudes can be considered (given the required 200 m separation between clouds and dust), whereas if the dust base is higher, even clouds at higher altitudes may be included. This could perhaps be significant if very low and slightly higher clouds responded differently to the effects of dust above.

   Thank you for pointing this out. We have added relevant discussion in Section 4, which reads.

   *"In addition to these caveats, limitations in our methodological approach may also introduce possible uncertainties in our estimates. First, while we chose only profiles having a minimum separation of 200 m between the cloud top and the dust-layer base to account for retrieval biases and avoid any microphysical interactions (see Section 2.2.1)., variabilities in the dust-layer base may result in associated variations in cloud-top height that may confound with the radiative impact of the dust layer. Specifically, this constraint means that when the dust base is low (e.g., 1 km), only profiles having clouds with very low top heights were allowed, potentially biasing the sample toward lower cloud altitudes. In contrast, while profiles having higher dust-layer base height can allow clouds having with*

*higher cloud tops. As a result, the observed cloud response to dust base height could be influenced by variations in cloud-top height across these categories. However, aside from the 1 km dust base category, cloud-top heights across other dust base height groups are broadly similar (see Table S1 and Fig S13), suggesting that differences in cloud top height are not the primary reason for the observed cloud response. In addition, the sensitivity analysis confirms that variations in cloud properties across the categories have minimal influence on the impact of dust geometric thickness and dust base height on dust-induced cloud-top heating (see Fig. 13 b and relevant discussion in Section 3.5.1)."*

6. Line 772: I wonder if "shortwave cooling" should be changed to "shortwave warming". If this was not the case, a clarification could help explain whether this refers to the shortwave impact of the dust layer, which is indeed cooling, or to the overall shortwave effect of sunlight, which is warming. (I think the latter, as sentence begins with "In contrast to the dust layer".)

Thank you for catching this. The term "shortwave cooling", in this context, refers to dust-induced shortwave cooling within the cloud layer, not the overall radiative effect of incoming solar radiation. We have clarified this point in the revised manuscript. The added sentence is following:

*"In contrast to the dust layer, dust-induced longwave warming within the low-level cloud dominates over the dust-induced shortwave cooling, which results from increased longwave downwelling radiation emitted by the dust layer and the reduction of solar radiation reaching the cloud top due to attenuation by the overlying dust. The net heating rate within the cloud also increases with the dust-layer geometric thickness (Fig. 8b and Fig. S9)."*

7. Lines 829-831: The finding that longwave warming increases with dust geometrical thickness seems puzzling to me, as a thicker dust layer means that some of the dust is at higher altitudes and is therefore cooler, emitting less longwave radiation towards the clouds below. It would help to include an explanation or at least a hypothesis about this. Could perhaps a relationship between dust geometric thickness and dust optical depth play a role? The same applies to Lines 837-838 and Lines 1100-1102.

We agree that further explanation is warranted. We have revised the manuscript to include the following:

*"Second, while we chose defined categories for dust geometric thickness, the variabilities in dust vertical distribution within each category may have influenced the associated radiative impacts on the low-level clouds. Specifically, despite a colder dust layer top, variabilities in the vertical distribution of coarse dust may have resulted in larger dust-induced longwave warming at cloud-top for thicker than thinner dust layers. For example, for the same dust-layer base height, maximum dust longwave extinction (unlike shortwave extinction) appears skewed to the bottom of a thicker dust layer than a thinner dust layer (e.g., Fig. S9 and S13), resulting in larger downwelling longwave radiation. With the inherent correlation between dust geometric thickness and dust optical depth (e.g., Fig.*

*S17), it is difficult to separate the resulting influence of co-variability in coarse dust vertical distribution, dust optical depth, and dust geometric thickness on the low-level cloud responses."*

8. Line 1030: The text says that "these factors have little impact on the cloud response to dust-layer characteristics", whereas Figure 14 only shows that these factors have little impact on dust characteristics but does not directly show the effect of dust on the clouds below. I guess the argument is that if these factors don't have a strong relationship with dust properties, they cannot affect dust influences on clouds either—but it would help to clarify this in the text.

We thank the reviewer for pointing this out. We have clarified the connection between meteorological variables and cloud responses. Our argument is that the weak sensitivity of key meteorological variables (e.g., Sea Surface Temperature, Relative Humidity and wind speed) to dust geometric thickness and dust base height (Fig 14) along with minimal and non-systematic variations (Fig S16) in these variables across categories suggests that these factors are unlikely to be responsible for the observed differences in cloud responses to dust-layer characteristics. We added a sentence in the revised manuscript which reads:

*"We find minimal variation in most cloud-controlling factors across our defined categories (Fig. S16), and their weak sensitivity to changes in dust geometric thickness and dust base height (Fig. 14) suggests limited influence of these factors on the estimated cloud response to dust-layer characteristics."*

**Minor wording issues**

1. Manuscript title: I think there should be no period at the end of the title. Also, as it is more typical to have titles that are not full sentences (with subject, verb, object, etc.) but are descriptors of the topic, I'd suggest rephrasing the title to something along the lines of "Dust semi-direct effects: The influence of dust-induced longwave radiation on low-level cloud response to free-tropospheric dust over the North Atlantic Ocean". This could help because the verb in the current wording, "influences", is a word that is sometimes a noun—and I needed to read the title several times to realize that here it is a verb.

Thank you. The title now reads *"Dust semi-direct effects: The influence of free-tropospheric dust-induced longwave radiation on low-level cloud response over the North Atlantic Ocean"*

2. Manuscript text and figures: The manuscript often uses three acronyms for dust characteristics: Equipped with an onboard lidar, this remote sensing platform measures the backscatter signal of clouds and aerosols at two wavelengths, 532 nm and 1024 nm, and with distinct vertical resolutions: 30 meters from the surface to 8.2 km, 60 m from ~8.2 km to ~20.2 km, and 180 m above ~20.2 km., GT, DOD for the base height, geometric thickness and optical depth of dust layers, respectively. For consistency, I recommend changing the notation to include the letter D even for dust geometric thickness (making it DGT), and to include the letter H representing "height" for dust base

height (making it DBH). This would treat all three quantities consistently, with three-letter acronyms representing three-word quantities (DBH, DGT, DOD). Even when full words are used, I recommend adding, for consistency, "dust" in front of "geometric thickness" (just as dust is mentioned for "dust base height" and "dust optical depth").

*Thank you. All instances of DB are converted to DBH and GT to DGT. In addition, labels in Fig 4, Fig 5, Fig 6, Fig 10, Fig13, and Fig14.*

3. Lines 35, 38, and 40: I suggest deleting the words "the".

   *Thank you. This is corrected now.*

4. Line 42: I suggest adding a comma after "interactions".

   *Thank you. This is corrected.*

5. Line 58: I recommend deleting the letter "s" from the end of "types".

   *Thank you. Modified*

6. Line 61: I recommend changing "In addition" to something like "On the other hand".

   *Thank you. We changed the sentence. It now reads*

   ***On the other hand,*** *when the biomass-burning aerosols are distinct*

7. Lines 65-66: The wording should be refined to clarify "far more substantial": More substantial than for what clouds? Perhaps changing "their influences on marine low-level clouds are far more substantial" to "their influences are by far the most substantial on marine low-level clouds".

   *Thank you. We modified the sentence*

   *Although semi-direct effects from biomass-burning aerosols occur for all cloud types, **their influences are by far the most substantial on marine low-level clouds***

8. Figure 1: For Figure 1c, it would help to clarify whether assuming that all aerosols are above the cloud layer implies that there is no aerosol inside or below the cloud layer. It should also be clarified whether the color bar for Figure 1d also applies to Figure 1c. The color bar should also have a label indicating what the colors represent. (The caption explains this a bit, but it should still be shown in the figure as well.)

   *Thank you, we have changed the Figure 1 legend and labelling on the colorbar.*

9. Line 77: It should be clarified what the word "Method" refers to—perhaps Section 2.2?

Thank you. The caption now reads (*see Section 2.2. ..*)

10. Line 102: I suggest changing "like" to "similarly to studies of".

Thank you we changed the mentioned sentence

*In addition, **similarly to studies of biomass-burning aerosol**s, these previous studies have linked the enhancement of low-level cloudiness and the associated negative dust SDE to the above-cloud dust layer to stronger lower-tropospheric inversion caused by dust-induced shortwave heating near the cloud top*

11. Line 130: I suggest deleting "the" after "of".

Thank you. Corrected.

12. Line 133: Because of the presence of "While" at the beginning of the sentence, the text ", however," should be deleted.

Thank you. Deleted.

13. Line 140: I recommend deleting the letter "s" from "aerosols".

Thank you. Corrected this.

*Specifically, we leveraged aerosol and cloud information from CALIOP..*

14. Line 171: I recommend adding "in" or "during" after "than".

Thank you for pointing out this now corrected sentence

*Specifically, during this period, there is a higher probability of dust being present above the clouds than **during** other periods of the year, allowing for enough data samples for statistically significant analysis*

15. Figure 2: In the legend of Panel a, a letter "s" should be added at the end of "month".

Thank you. Specifically, Figure 2 is modified to include the suggestion

16. Line 168: I suggest changing "retrievals" to "operational data products".

Thank you now phrase changed to *"CALIOP level 2 data products."*

17. Line 170: The word "is" should be added before or after "typically".

Thank you. We changed the sentence; now it reads,

*We focused our analysis on the North Atlantic Ocean between May and August (2007-2017) when the SAL dust frequently overlies low-level cloud layers*

18. Line 183: I recommend changing "aerosols, clouds, and radiative fluxes" to "aerosol, cloud, and radiative flux". I also recommend adding a comma after "observations".

    Thank you. We changed the sentence

    *We obtained aerosol, cloud, and radiative flux information from satellite-based observations, and meteorological parameters, including temperature and humidity, from reanalysis datasets.*

19. Line 186: I recommend either adding "the" in front of CALIPSO, or removing "satellite".

    Thank you, included. *The satellite-based observations include those from CALIOP (Cloud-Aerosol Lidar with Orthogonal Polarization) onboard **the** CALIPSO (Cloud-Aerosol Lidar and Infrared Pathfinder Satellite Observation) satellite, cloud profiling radar (CPR) aboard CloudSat, and reanalysis data from the European Centre for Medium-Range Weather Forecasts (ECMWF).*

20. Line 189: The letter "s" should be deleted from the end of "Aerosols".

    Thank you, changed. *We relied on CALIOP for vertically resolved measurements of cloud and aerosol (Winker et al., 2010).*

21. Line 191: I recommend deleting the hyphen between "remote" and "sensing".

    *Equipped with an onboard lidar, **this remote sensing** platform measures the backscatter signal of clouds and aerosols at two wavelengths, 532 nm and 1024 nm, and with distinct vertical resolutions: 30 meters from the surface to 8.2 km, 60 m from ~8.2 km to ~20.2 km, and 180 m above ~20.2 km.*

22. Line 193: I recommend changing the symbol "~"to "≈" all three times (and throughout the manuscript).

    Thank you for the suggestion. Corrected for all instances throughout the manuscript

23. Lines 196, 230, 241, and throughout the manuscript: There should be a space in "532nm".

    Thank you for the suggestion. Corrected for all instances throughout the manuscript

24. Line 202" I recommend adding a letter "s" at the end of "measurement".

    Thank you. Included in the sentence. *Compared to ground-based lidar **measurements**,*

25. Line 206: I recommend adding "the" or "this" after "improved".

   Thank you. Corrected. *The latest release has improved **this** issue*

26. Lines 259-260: Either a "the" should be added after "on", or "satellites" should be deleted.

   Thank you. Added, part of sentence is following

   *leverage instruments on **the** CALIPSO and CloudSat satellites.*

27. Line 263: I recommend adding "the" after "from".

   Thank you. Corrected.

   *includes measurements from **the** CPR aboard CloudSat,*

28. Line 374: I recommend replacing "confound" by "be confounded" or by "complicated".

   Thank you.

   *Previous studies have shown that meteorology **may be confounded** with aerosol-cloud interactions,*

29. Line 380: The word "degree" could be replaced by the degree sign, as in Line 384 and elsewhere.

   Thank you.

   *divide our region of interest into 5˚×5˚ grid boxes.*

30. Lines 407-408: As is, f_cat is singular and f_lowcld is plural; consistency should be achieved by changing one or the other.

   Thank you. Corrected both changed to "***is***"

31. Line 419: I recommend changing the word "study" to "paper" or "manuscript".

   Thank you. Now sentence reads,

   *In addition, although not included **in this paper**, we also analyzed other similar properties, such as cloud optical depth, for each category, with similar conclusions to cloud fraction.*

32. Line 481 and elsewhere as needed: I recommend changing "net" to "total", as net would imply the difference (and not the sum) of two quantities. If it is the difference, though, that should be explained.

*Thank you. Corrected at each instance.*

33. Line 526: A comma should be added after "Subsidence".

    *Thank you. Corrected the sentenced*

    *Subsidence, often quantified using vertical velocity at 700 hPa or adjacent layers above the cloud*

34. Line 554 and elsewhere in the manuscript: The letter "s" in "section" should be capitalized whenever it is for a specific section identified by its number (e.g., Section 3.1).

    *Thank you. Included the suggestion.*

35. Line 708: For clarity, the word "the" should be changed to "increases in all three considered".

    *Thank you. Corrected it now the sentence reads*

    *Overall, our results, examining the effects of dust base heights, dust geometric thicknesses, and dust optical depth on low-level cloud cover, indicate that low-level clouds respond negatively to increase in **all three considered** dust properties and characteristics, but more strongly to increases in dust optical depth and geometric thickness, than to the same increases in dust base height.*

36. Lines 720 and 721: The hyphens could be deleted from "dust-layer" and "dust-base", for consistency with the rest of the manuscript.

    *Thank you. Corrected the sentence*

    *Within this context, therefore, we seek to understand the radiative influence of **dust layer** characteristics on low-level cloudiness. To do so, we examined how changes in **dust base heights, dust geometric thickness,** and dust optical depth influence radiative heating rates within the dust layer and at the underlying low-level cloud top, and consequently, the relationship with the low-level cloudiness.*

37. Line 746: The last digit ("3") should be deleted from "approximately 0.723 km".

    *Thank you changed it to "approximately 0.72 km"*

38. Lines 773 and 774: The "s" should be deleted from the end of "rates".

    *Thank you. "the **heating rate** also increases with the dust-layer geometric thickness.*

    *the increase in **heating rate** ranges between ≈ 0.2"*

39. Lines 819-820: For clarity, I recommend changing "change" to "increase", and adding "increased" after "pattern of" (if this is correct). A similar clarification is also needed in Line 1104.

*Thank you for the suggestion now modified in manuscript "As a result, we calculated the heating rate response of the dust-induced cloud-top radiative warming to a unit **increase in DOD** and found a consistent **pattern of increased dust-induced** warming for all combinations of dust-base height and geometrical thickness."*

40. Line 825: I recommend deleting the word "the".

*dependence of cloud-top longwave radiative heating changes on dust-layer characteristics,*

41. Line 857: I recommend changing "on" to "to".

*Thank you changed the sentence. "The contributions of dust optical depth **to** cloud top dust-induced longwave warming are about 8 times compared to the contribution due to dust geometric thickness,"*

42. Lines 858 and 859: I recommend adding "larger" after "times", if this is correct.

*Thank you included in the sentence. "The contributions of dust optical depth to cloud top dust-induced longwave warming are about **8 times larger** compared to the contribution due to dust geometric thickness, and about **6 times larger** compared to the contribution due to dust base height."*

43. Line 865: I recommend adding "the contribution of" after "than". I also recommend changing "sensitively" to "sensitivity".

*Thank you we changed the sentence "While the contribution of dust optical depth to the dust-induced cloud-top longwave warming response is stronger than **the contribution** of other dust-layer characteristics (e.g., Fig. 10), the impact of dust optical depth **sensitivity** depends on dust physicochemical and absorption properties, including the dust size distribution and dust complex refractive indices."*

44. Line 883: I recommend changing "proportion" to "fraction".

*Thank you. We changed it to*

*The coarse-mode **fraction** decreases from around 89% near the source to approximately 50% for long-range transport.*

45. Line 958: I recommend inserting the words "influence of" after "from the".

*Thank you included the word*

*We use SBDART to separate the influence of the background thermodynamic profile, including the influence of temperature and humidity, from the **influence** of dust properties, and examine their contribution to the simulated radiative heating rates.*

46. Line 980: The caption should clarify what kind of altitude is along the right side y axis: Is this perhaps cloud top height or dust layer top or thickness?

    This is altitude where lower level of bars like previous plots e.g. Fig 3, from dust layer base to dust layer top, we have not plotted the cloud-top height in this Fig.

47. Lines 1059, 1065, and 1066: I guess "LST" should be changed to "LTS".

    Thank you for pointing this. Now changed LST to LTS

48. Line 1069: The word "sensitively" should be deleted, or the word "depends sensitively on" should be replaced by "is sensitive to".

    Thank you. *We have shown that low-level cloud response **is sensitive to** the dust-layer altitude, geometric thickness, and dust optical depth, influencing the dust-induced longwave warming and counteracting the mean radiative cooling near the low-level cloud top.*

49. Line 1074: I recommend inserting "simulations for the period" at the beginning of the line.

    Thank you. Included.

50. Line 1089: My sense is that the word "relative" should be deleted, as the paper examines the absolute altitude of dust layer base above sea level, but does not examine the separation between cloud and aerosol layers (which would be a "relative altitude").

    Thank you. Deleted the word relative,
    *"we find that the extent of such increases depends on the dust layer's optical properties as well as its altitude and geometrical thickness."*

51. Line 1234: There should be a space after "clouds".

    Thank you. Corrected.

52. Line 1257: I recommend replacing ", especially studies found the" by ". Most importantly, studies found an".

    Thank you. Corrected. *"leading to uncertainties in the estimation of dust's SDE (Cappa et al., 2016), **most importantly, studies found an underestimation of large dust particles by** current climate models, which may lead to potential underestimation of the longwave effect"*